# Seasonal isoprene emission estimates over tropical South America inferred from satellite observations of isoprene

Shihan Sun[1,2], Paul I. Palmer[1,2], Richard Siddans[3,4], Brian J. Kerridge[3,4], Lucy Ventress[3,4], Achim Edtbauer[5], Akima Ringsdorf[5], Eva Y. Pfannerstill[5], Jonathan Williams [5,6]

[1]National Centre for Earth Observation, University of Edinburgh, Edinburgh, EH9 3FF, UK
[2]School of GeoSciences, University of Edinburgh, Edinburgh, EH9 3FF, UK
[3]Remote Sensing Group, STFC Rutherford Appleton Laboratory, Chilton, Oxfordshire, OX11 0QX, UK
[4]National Centre for Earth Observation, STFC Rutherford Appleton Laboratory, Chilton, Oxfordshire, OX11 0QX, UK
[5]Max Planck Institute for Chemistry, Atmospheric chemistry department, Mainz, Germany
[6]Climate and Atmosphere Research Center (CARE-C), The Cyprus Institute, 1070 Nicosia, Cyprus

*Correspondence to*: Susie.Sun@ed.ac.uk

**Abstract.** Isoprene, a volatile organic compound (VOC) emitted by plants, plays a significant role in atmospheric chemistry and climate. The Amazon rainforest is a globally-relevant source of atmospheric isoprene. We report isoprene emissions inferred from a full-physics retrieval of isoprene columns from the Cross-track Infrared Sounder (CrIS) and the local sensitivities between isoprene emissions and isoprene columns determined by the GEOS-Chem chemical transport model. Compared with the MEGAN isoprene emissions, the isoprene emission estimates inferred from CrIS have different spatial and seasonal distributions with generally lower emission rates but with higher emission rates over the northern Amazon basin and southeast Brazil. The observed mean isoprene concentration at the Amazon Tall Tower Observatory (ATTO), March—December 2019, is 3.0 ± 2.2 ppbv, which is reproduced better by the GEOS-Chem model driven by isoprene emissions inferred from CrIS (2.8 ± 1.4 ppbv) than by the MEGAN inventory (4.1±1.3 ppbv). Isoprene emission estimates inferred from CrIS generally agree better than MEGAN with *in situ* observations of seasonal isoprene fluxes over the Amazon. GEOS-Chem model formaldehyde (HCHO) columns, corresponding to isoprene emissions inferred from CrIS, are generally more consistent with TROPOMI data (normalized mean error, NME = 43%) than the HCHO columns corresponding to MEGAN isoprene emissions (NME = 50%), as expected. CrIS inferred isoprene emission rates can vary by ± 20% considering potential model biases in nitrogen oxide emissions. Our results provide confidence that we can use CrIS data to examine future impacts of anthropogenic activities on isoprene emissions from the Amazon.

## 1 Introduction

Tropical South America, including the Amazon rainforest, hosts important ecosystems that influence the global carbon and water cycles. Amazonia is also a significant but uncertain source of biogenic volatile compounds (BVOCs), dominated by mass by isoprene (Guenther et al., 2006), that influence atmospheric chemical composition on local to global scales

(Wiedinmyer et al., 2006; Yáñez-Serrano et al., 2015; Millet et al., 2016; Gomes Alves et al., 2023; Mayhew et al., 2023; Ringsdorf et al., 2023; Ferracci et al., 2024). Isoprene has an e-folding lifetime of ~1 hour against oxidation by hydroxyl radical (OH) and plays a role in ozone chemistry (Atkinson, 2000; Saunier et al., 2020), production of secondary organic aerosol (Claeys et al., 2004; Kroll et al., 2005, 2006; Carlton et al., 2009), and by modifying the levels of OH (Lelieveld et al., 2008; Hofzumahaus et al., 2009; Fuchs et al., 2013; Millet et al., 2016; Nölscher et al., 2016; Hansen et al., 2017; Pfannerstill et al., 2021). Isoprene also influences the lifetimes of other pollutants, e.g. carbon monoxide (Miyoshi et al., 1994) and methane (Feng et al., 2023). Evidence suggests that plants emit isoprene to protect leaf biochemistry under environmental stress (Sharkey et al., 2007; Monson et al., 2013; Zeinali et al., 2016), which is also seen as a key plant trait that determines species responses to rising temperature and drought (Taylor et al., 2018; Werner et al., 2021; Byron et al., 2022). Changes in isoprene emissions due to deforestation, rising levels of atmospheric $CO_2$, and climate change induced land use and land cover changes will also play an important role in controlling future changes in biogenic emissions and thereby atmospheric composition (Fini et al., 2017; Chen et al., 2018; Yáñez-Serrano et al., 2020; Sahu et al., 2023).

Isoprene emissions are typically described in atmospheric chemistry transport models and chemistry-climate models using a bottom-up models. The Model of Emissions of Gases and Aerosols form Nature (MEGAN) (Guenther et al., 2006, 2012)  is a commonly-used bottom-up emission model, which we use in this study, so it is instructive to use it to explain the general approach. Basal emission rates, indicative of standard environmental conditions and specific genera, are grouped into a comparatively small set of plant functional types (PFT) that describe a group of plants with similar characteristics. These standardised emission rates are then adjusted using empirical scaling factors that describe environmental changes, e.g., temperature, photosynthetic active radiation. Large uncertainties remain for these basal isoprene emission rates (Arneth et al., 2008) and for the empirical parameterizations of how different plants respond to their environment (Jiang et al., 2018; Seco et al., 2022; Bourtsoukidis et al., 2024). As such, these uncertainties compounded with other uncertainties with time-dependent inputs, e.g., landcover, can compromise model performance. Uncertainties from leaf-level phenological traits, such as leaf age or ecosystem-level plant biodiversity, which also influence isoprene emissions are difficult to measure, with very few observational sites, but can be partially addressed with satellite-based observations of optical wavelengths (Li et al., 2024; Lian et al., 2024). Limitations of satellite remote sensing data results in uncertainties in the inferred maps of vegetation, including coarse resolution and PFTs mapping, which subsequently introduces uncertainties to isoprene emission estimates (Chen et al., 2018; Opacka et al., 2021). Direct evaluation of these emission models is difficult, particularly over tropical ecosystems where there are very few data. An alternative indirect approach is to compare the emissions model with atmospheric data in which an atmospheric chemistry transport model acts an intermediary between the emissions and the corresponding atmospheric concentrations of isoprene and its oxidation products, e.g. formaldehyde (HCHO). Studies have reported significant discrepancies between MEGAN isoprene emission estimates, used as an input to a chemistry transport model, and observations of atmospheric isoprene determined by *in situ* and satellite data, particularly over tropical ecosystems (Warneke et al., 2010; Bauwens et al., 2016; Wang et al., 2017; Gomes Alves et al., 2023).

Satellite observations of HCHO have for the last 20 years helped to supplement sparse ground-based observations of isoprene (Abbot et al., 2003; Palmer et al., 2003; Shim et al., 2005; Wiedinmyer et al., 2005; Palmer et al., 2006, 2007; Barkley et al., 2008; Millet et al., 2008; Müller et al., 2008; Barkley et al., 2009; Stavrakou et al., 2009; Kaiser et al., 2018; Surl et al., 2018; Opacka et al., 2021; Feng et al., 2024; Opacka et al., 2024). Formaldehyde is a high yield product of isoprene oxidation by OH and because HCHO has a lifetime of typically only several hours, observed changes in HCHO can be linked to emissions of the parent hydrocarbon (Palmer et al., 2003). Irrespective of the sophistication of the inverse method that is used to translate observed changes in HCHO to isoprene emission estimates (Shim et al., 2005; Kaiser et al., 2018) some form of atmospheric chemistry model is needed, typically a global 3-D model that includes atmospheric transport, so we remain reliant on the assumed *a priori* emission inventories (e.g. Guenther et al, 2006), the chemical mechanism and their respective uncertainties. For example, there remain substantial uncertainties associated with the production of HCHO from isoprene oxidation at low nitrogen oxide levels, which are found over the tropics away from biomass burning and urban centres (Lelieveld et al., 2008). Interpreting HCHO in terms of BVOC emissions also requires careful attention to discard data influenced by biomass burning (Barkley et al., 2008; Gonzi et al., 2011). HCHO can also be produced from sources other than isoprene, such as alkanes, alkenes, and monoterpene, resulting in uncertainties in HCHO inferred isoprene emissions (Marvin et al., 2017; Surl et al., 2018). Some studies have highlighted a positive model bias for MEGAN over the tropics compared with isoprene emission estimates inferred from satellite observations of HCHO (Marais et al., 2012; Barkley et al., 2013; Stavrakou et al., 2014; Worden et al., 2019), while others have found a negative bias when compared with ground based or aircraft measurements (Gu et al., 2017; DiMaria et al., 2023). Development of isoprene retrievals (Fu et al., 2019; Palmer et al., 2022; Wells et al., 2020, 2022) using data collected by the Cross-track Infrared Sounder (CrIS) has resulted in a new and independent capability to determine isoprene emissions more directly. Different approaches have been adopted to retrieve isoprene from CrIS data. Fu et al. (2019) developed the first direct retrieval of isoprene using infrared radiance measurements from CrIS, using the MUSES algorithm which follows optimal estimation principles, while others have adopted other optimal estimation retrieval approaches (Palmer et al., 2022). In more recent work, others have developed an innovative machine learning approach (Wells et al, 2022), building on Fu et al. (2019). Here we use data retrieved using optimal estimation.

In this study, we use a nested version of the GEOS-Chem chemical transport model to investigate the consistency of isoprene emission estimates inferred from CrIS isoprene retrievals and the bottom-up MEGAN isoprene emission inventory over tropical South America during 2019, and compare the *a posteriori* isoprene concentrations with *in situ* measurements collected from the Amazon Tall Tower Observatory (ATTO), located in the pristine rainforest. Section 2 describes the nested GEOS-Chem model configuration we use to interpret the satellite and *in situ* tall tower data, including the MEGAN isoprene emissions model; the TROPOMI and CrIS satellite data and the tall tower data; and the methods we use to translate the column data into emission estimates. In Sect. 3, we report our results. We compare our satellite-based emission estimates for

100     isoprene with the inventory estimates from the MEGAN model and evaluate our CrIS derived estimates against the *in situ* tall tower atmospheric isoprene measurements. We also evaluate the CrIS derived isoprene emission estimates by comparing the corresponding model HCHO columns simulated with HCHO columns retrieved from TROPOMI, using the GEOS-Chem model as an intermediary. To examine the robustness of our isoprene emission estimates from CrIS data, we report the results from a series of sensitivity tests that use assume different soil $NO_x$ emission rates. We conclude our study in Sect. 4.

## 2 Data and Methods

Here we describe the GEOS-Chem atmospheric chemistry model that relates surface emissions of BVOCs, including isoprene, and atmospheric columns of isoprene and HCHO. We describe the satellite observations of isoprene from CrIS and HCHO from TROPOMI, and the tall tower measurements of atmospheric isoprene collected in central Amazonia that we use to evaluate the model. We also describe the methods that we use to translate these data into estimates of isoprene emission.

### 2.1 GEOS-Chem simulations

We use GEOS-Chem v14.1.0 atmospheric chemical transport model (https://geoschem.github.io, last access: 5 Dec 2024). GEOS-Chem is driven by Goddard Earth Observing System-forward processing (GEOS-FP) assimilated meteorological analyses from the NASA Global Modelling and Assimilation Office at NASA Goddard Earth Observing System. Nested model simulations are conducted at a horizontal resolution of 0.25° × 0.3125° using 47 vertical levels, of which 30-35 are within the troposphere, over a spatial domain centred over tropical South America: 35° S—15° N, 85°W—30° W. A buffer zone of 3° is applied along each of the four lateral boundaries of the nested domain. We generate lateral boundary conditions for the nested model using a self-consistent global model run at a horizontal resolution of 2° × 2.5°, following a one year spin-up from Jan 2018 through December 2019.

We use the complex secondary organic aerosol (SOA) and semi-volatile primary organic aerosol (SVPOA) mechanism, which includes the full-chemistry "tropchem" mechanism to describe gas-phase reactions (Eastham et al., 2014) and the photochemical production of SOA and SVPOA with up-to-date isoprene mechanisms (Bates and Jacob, 2019). The "complex–SOA_SVPOA" mechanism uses a combination of explicit aqueous uptake mechanisms (Marais et al., 2016) with a standard volatility basis set scheme (Pye et al., 2010).

We use the standard Harvard-NASA Emissions Component (HEMCO) configuration, including biogenic emissions from the MEGAN v2.1 inventory (Guenther et al., 2012). To test the isoprene emissions inferred from the satellite data, we use offline BVOC emissions at 0.25° × 0.3125° horizontal resolution which are pre-computed using MEGAN v2.1 using LAI estimates from the MODerate-resolution Imaging Spectroradiometer (MODIS) (Yuan et al., 2011) and GEOS-FP meteorological reanalyses to modify the emission rates. MEGAN uses an empirical $CO_2$ inhibition scheme to calculate isoprene emission

factors (Possell and Hewitt, 2011; Tai et al., 2013). The MEGAN extension in HEMCO does not include soil moisture effect for isoprene, so our current model configuration does not account for the impact of drought on isoprene emission. Pyrogenic emissions are from the Global Fire Emissions Database version 4.1 that includes small fire correction (Van Der Werf et al., 2017). The GFED inventory provides monthly dry matter emissions based on satellite observations of fire activity and vegetation coverage from MODIS. Anthropogenic emissions, including fossil and biofuel sources, are from the Community Emissions Data System inventory (CEDS v2), which provides CMIP6 historical anthropogenic emissions data from 1750 to 2019 mapped to a 0.5° global grid (Hoesly et al., 2018). Offline soil $NO_x$ emission estimates used in this study (Hudman et al., 2012) are generated using consistent GEOS-FP meteorological analyses.

To compare model simulations with satellite retrievals, GEOS-Chem simulated profiles are sampled at the satellite overpassing time and location of each measurement for both TROPOMI and CrIS. We then interpolate model profiles to the vertical levels of satellite retrievals. For consistency between satellite and GEOS-Chem simulated vertical profiles, we also apply scene-dependent averaging kernels that describe the instrument vertical sensitivity to changes in a trace gas, replacing any *a priori* information assumed by the retrieval, and then integrate from the surface up to the tropopause to calculate column values.

## 2.2 CrIS isoprene retrievals

We use CrIS isoprene column average retrievals from RAL's Infrared and Microwave Sounding (IMS) scheme (Palmer et al., 2022). CrIS is a Fourier transform spectrometer covering three IR spectral regions spanning 650–2550 cm[-1] launched onboard the Suomi-National Polar-orbiting Partnership (S-NPP) satellite in October 2011, NOAA-20 in November 2017, and NOAA-21 in 2022 into sun-synchronous low Earth orbits with overpass times of 01:30 and 13:30 local time. CrIS has comparatively low noise in the spectral region in which isoprene features occur which, together with more favourable thermal structure at ~13:30 than 9:30 make detection of isoprene feasible for CrIS. Other than instrumental noise, uncertainty in CrIS retrieved isoprene column averages principally concerns the adopted vertical profile shape, which is a constant volume mixing ratio, and CrIS vertical sensitivity, which is accounted for explicitly in the analysis by applying vertical averaging kernels to the model profiles. Although the *a priori* constraint on the retrieval is weak, this is also accounted for. As in Palmer et al (2022), CrIS isoprene data are filtered to exclude scenes with extensive thick or high cloud and retrievals with a high cost function (i.e., poor spectral fit). Due to the simple, adopted profile shape and decrease in sensitivity near surface level in absence of significant surface-air temperature contrast, IMS column averages tend to be lower than those derived from surface-based observations where surface level concentrations are high as expected, which does not necessarily indicate a low bias from RAL IMS product. The sensitivity of infrared spectra to trace gases is generally lowest near the ground because of the small temperature difference between the atmosphere and the surface, particularly at night. In this study, we use daytime satellite retrieved isoprene columns which correspond with peak isoprene emissions.

**2.3 TROPOMI column retrievals of HCHO and NO₂**

TROPOMI was launched onboard of the Copernicus Sentinel-5 Precursor (S5P) satellite on 13 October 2017 into a low-Earth polar orbit with an equatorial local overpass time of 13:30 (Veefkind et al., 2012). TROPOMI is a nadir viewing instrument that collects data at ultraviolet, visible, near infrared, and shortwave infrared wavelengths. TROPOMI has a horizontal swath of 2600 km that is divided into 450 across-track rows. The spatial resolution of TROPOMI at nadir is $3.5 \times 7$ km² (across-track × along-track) which was later refined to $3.5 \times 5.5$ km² in August 2019 due to an adjustment to the along track integration time. TROPOMI $NO_2$ retrievals use wavelengths from 400 to 496 nm and HCHO retrievals using wavelengths from 320 to 405nm. We refer the reader to dedicated reported on these retrieved data products for further details (De Smedt et al., 2018; Van Geffen et al., 2022). We use the operational offline TROPOMI level 2 quality control retrievals for HCHO and $NO_2$ columns. To remove retrievals with substantial errors or those influenced by clouds or snow/ice cover we use the retrieval quality assurance (QA) flag provided by the data products. We discard data with QA flags > 0.75 for $NO_2$ and > 0.5 for HCHO, following recommendations (De Smedt et al., 2020; Eskes and Eichmann, 2022)

TROPOMI has a better signal-to-noise ratio compared to Ozone Monitoring Instrument (OMI) but the HCHO observations still have a bias against ground-based multi-axis differential optical absorption spectroscopy instruments (De Smedt et al., 2021). TROPOMI retrievals of HCHO were found to underestimate high columns and overestimate low columns in previous studies (Vigouroux et al., 2020; Müller et al., 2024). Noting that biases for OMI and TROPOMI HCHO columns are expected to be similar (De Smedt et al., 2021), we have applied a bias-correction formula recently proposed for HCHO columns from OMI, which has previously been evaluated using observations over South America (Müller et al., 2024): $\Omega_{HCHO,BC} = (\Omega_{HCHO} - 2.5 \times 10^{15})/0.655$, where $\Omega_{HCHO,BC}$ denotes the bias-corrected HCHO columns (molec cm⁻²). We find that applying this bias does not change the conclusions of our paper.

**2.4 Amazon Tall Tower Observatory (ATTO)**

We use data collected at the Amazon Tall Tower Observatory (ATTO, 2°8'S, 59°0'W) site located in central Amazonia (Gomes Alves et al., 2023) to independently evaluate the GEOS-Chem model. The characteristics of this site have been described extensively in Andreae et al. (2015). The anthropogenic influence from the closest city Manaus (150 km southwest of ATTO) is negligible and the site has been established to represent pristine tropical forest conditions throughout the year. The tropical climate at this broader geographical region includes a dry season (July – October) and a wet season (December – May) associated with seasonal rainfall amounts of less than 100 mm and over 200 mm, respectively (Botía et al., 2022). We use air measurements that were collected at 80m, 150m and 320m respectively from March to December 2019. The measurements were made using a Proton Transfer Reaction Time of Flight Mass Spectrometer (PTR-ToF-MS Ionicon Austria) as described by Ringsdorf et al. (2023). For the purposes of comparison with the model, we calculate the mean hourly observed isoprene concentrations from these three levels. Modelled isoprene mixing ratios from the corresponding

first three levels from the surface are sampled at the day and time when observations are available, and then averaged over the same time period.

**2.5 Method to infer satellite-derived isoprene emission inventory**

We use CrIS retrievals of isoprene column ($\Omega_{isop}$), described above, to derive the isoprene emission rates that we use within the GEOS-Chem model. To understand the relationships of these isoprene columns to isoprene emissions, we use a linear

model to regress MEGAN isoprene emission rates $E_{isop,GC}$ (kg m$^{-2}$ s$^{-1}$) and the corresponding GEOS-Chem model $\Omega_{isop}$. To determine monthly isoprene emission rates from satellite retrievals from CrIS, we rearrange the regression model and insert the observed columns: $E_{isop,sat} = (\Omega_{sat} - B)/S$, where $E_{isop,sat}$ is the isoprene emission estimate inferred by satellite data, $\Omega_{sat}$ refers to the CrIS column data. The intercept $B$ refers to the isoprene background, while the slope $S$ refers to the isoprene column corresponding to the isoprene emission rates which is mainly determined by isoprene lifetime. We use a similar

approach to relate TROPOMI HCHO columns ($\Omega_{HCHO}$) to isoprene emission estimates. For the analysis of HCHO data, the $S$ in the regression model is determined by the HCHO yield from isoprene oxidation and by the HCHO lifetime.

We first compute the linear regression relationships within each grid for each month using daily MEGAN isoprene emission estimates and the corresponding model values for $\Omega_{HCHO}$ and $\Omega_{isop}$ sampled at the equatorial overpass time of the satellite,

2018—2020, inclusively. We then use these grid-based regressions models to infer monthly isoprene emission estimates for 2019. For model grid boxes for which emission rates cannot be estimated, e.g., p-value > 0.05 or missing data, we use data from the immediately adjacent grids (nearest neighbours) to recalculate the regression models, as described above. The magnitude of satellite inferred isoprene emission rates for 2019 is scaled by the ratio of monthly MEGAN emission rates in 2019 relative to the 2018-2020 monthly mean. We then relate the monthly $E_{isop,sat}$ values, representative of the approximate

13:30 local overpass time of CrIS and TROPOMI, to the diurnal variation in isoprene emission rates by using scaling factors derived from diurnal and day-to-day variations in the offline MEGAN isoprene emission rates for 2019. Given the lifetime of isoprene against oxidation by OH and the mean wind speed we estimate that most of the isoprene lost and the associated HCHO production is on a scale shorter than a 2° × 2.5° grid box but typically longer than an individual 0.25° × 0.3125° grid box. Consequently, to remove this potential "smearing effect" on the finer horizontal resolution, we calculate our regression

model using a horizontal resolution of 2° × 2.5°, following recent studies (Wells et al., 2020, 2022).

To remove the influence of biomass burning on the HCHO regressions models, we discard data for which there are fire counts identified by the NASA Fire Information for Resource Management System (FIRMS) active daily fire data acquired by the MODIS sensors (https://firms.modaps.eosdis.nasa.gov/, last access: 15 Nov, 2024). These fire counts are determined

by thermal IR anomalies by the MODIS sensors aboard Aqua and Terra satellites at a 1km horizontal resolution. We select

daytime fire counts with a high confidence level, i.e., higher than or equal to 80% as recommended in the MODIS user's guide (Giglio et al., 2020).

## 3 Results and discussion

### 3.1 CrIS inferred isoprene emissions

Figure 1 compares monthly mean CrIS and GEOS-Chem (MEGAN) isoprene columns for year 2019. GEOS-Chem model fields are screened where CrIS data are absent or do not pass the quality thresholds. Both GEOS-Chem and CrIS show a strong seasonal cycle, with a peak monthly mean $\Omega_{isop}$ in August. GEOS-Chem (MEGAN) and CrIS report the lowest monthly $\Omega_{isop}$ in April and November, respectively. The best agreement between GEOS-Chem (MEGAN) and CrIS for 2019 is found mainly during dry months, from June to August, with Pearson correlation coefficients R=0.59—0.73, $p < 0.05$, and

with normalized mean biases (NMB) of 20% to 38%. The largest discrepancies between GEOS-Chem and CrIS typically occur during relatively wet months (regional mean total precipitation > 5mm day$^{-1}$). GEOS-Chem (MEGAN) has a positive bias with respect to CrIS (NMB > 100%) over the Amazon Basin throughout the year with the highest positive biases over the western Amazon basin as shown in Fig.1. Despite the overall positive biases, the model underestimates $\Omega_{isop}$ over southeast Brazil where it is dominated by savanna, with the largest negative biases during the dry season. These seasonal and

regional model biases have also been found in previous studies (e.g., Wells et al., 2020). Hotspots of $\Omega_{isop}$ during the wet season are mainly observed to the north of the Amazon basin, on the borders between Columbia, Venezuela, and Brazil, where the land cover is dominated by tropical rainforest. In contrast, GEOS-Chem (MEGAN) shows regional hotspots along the east of the Andes, over western Brazil, and eastern Peru. Elevated values of CrIS $\Omega_{isop}$ over northern Amazonia has been independently observed by aircraft measurements (Gu et al., 2017), suggesting possible negative model bias where the

tropical plant species distributions may not be well represented by the model.

Previous studies have reported significant spatial differences between bottom-up emission inventories of isoprene and satellite column observations of isoprene (Fu et al., 2019; Wells et al., 2022; J.-F. Müller et al., 2024). To understand these differences, we calculate top-down values of $E_{isop}$ using satellite column retrievals of isoprene and HCHO to compare with

the MEGAN bottom-up inventory for $E_{isop}$. We use the relationships between isoprene emissions and isoprene columns described in GEOS-Chem to derive $E_{isop}$ from CrIS $\Omega_{isop}$, and we also calculate $E_{isop}$ derived from TROPOMI $\Omega_{HCHO}$ to compare with CrIS derived $E_{isop}$. For those model grids where the $E_{isop} \sim \Omega_{sat}$ linear relationships are not significant (p-value > 0.05), or satellite inferred isoprene emissions are negative, we assume no isoprene emissions within these grids so that any differences in modelled biases are caused by CrIS derived $E_{isop}$ that have been modified in this study. This approach can

cause an underestimation of $E_{isop}$ in some areas. Figure 1 also shows the monthly location of fires identified by MODIS data. It clearly shows that for large parts of the Amazon, isoprene emission estimates inferred from HCHO are compromised by

fire (Barkley et al., 2008, 2011) and for these locations we remove days with fire incidents when computing $E_{isop} \sim \Omega_{sat}$ linear relationships.

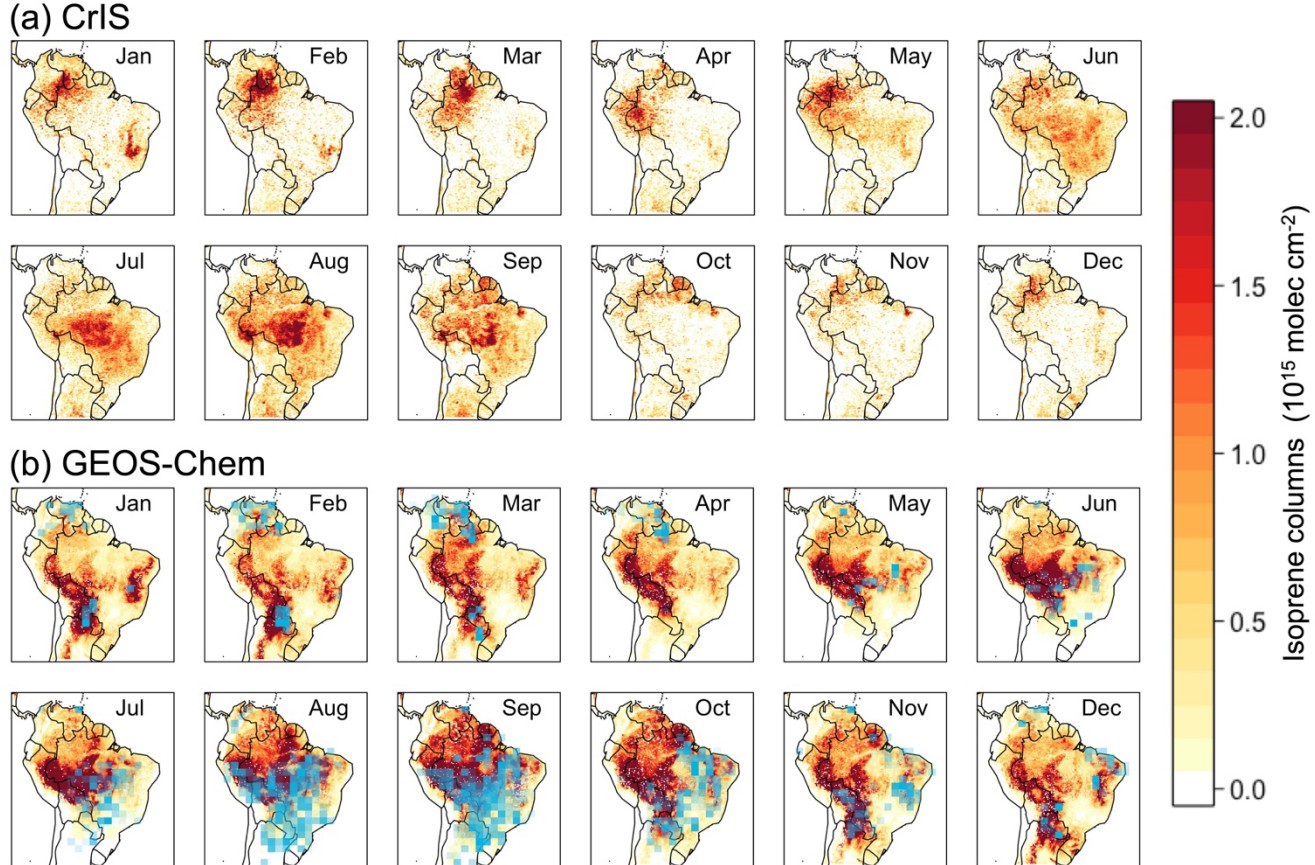

**Figure 1: Monthly mean (a) CrIS and (b) GEOS-Chem isoprene columns ($10^{15}$ molec cm$^{-2}$) driven by MEGAN emissions sampled at the CrIS local overpass time of 13:30 for 2019. GEOS-Chem model columns include scene-dependent CrIS averaging kernels. Blue boxes in (b) indicate fire intensities and locations from MODIS.**

Figure 2a shows monthly total isoprene emission estimates from MEGAN and with estimates inferred from CrIS isoprene and TROPOMI HCHO column data over the spatial domain showed in Fig. 1 where monthly isoprene emission rates can be inferred from both CrIS and TROPOMI data. TROPOMI and CrIS inferred isoprene emission estimates peak in September, same with MEGAN. TROPOMI derived isoprene emission estimates are 12~72% lower than MEGAN. CrIS is about 2~49% lower than MEGAN except for July. We remove $\Omega_{HCHO}$ values that coincide with MODIS detected fires, resulting in lower TROPOMI $\Omega_{HCHO}$ derived isoprene emissions during the months where isoprene hotspots are collocated with fire incidents as shown in Fig. 1. Recent work also found that OMI HCHO based isoprene emissions were significantly lower than the CrIS-derived emissions (from a different CrIS retrieval scheme) over South America, with the largest discrepancies over

Brazil (Müller et al, 2024). Figure 2b shows the spatial distribution of monthly mean isoprene emission rates for year 2019, inferred from the IMS CrIS isoprene column data. The corresponding monthly isoprene emission estimates from MEGAN and inferred from TROPOMI HCHO columns are shown in Fig. S1.

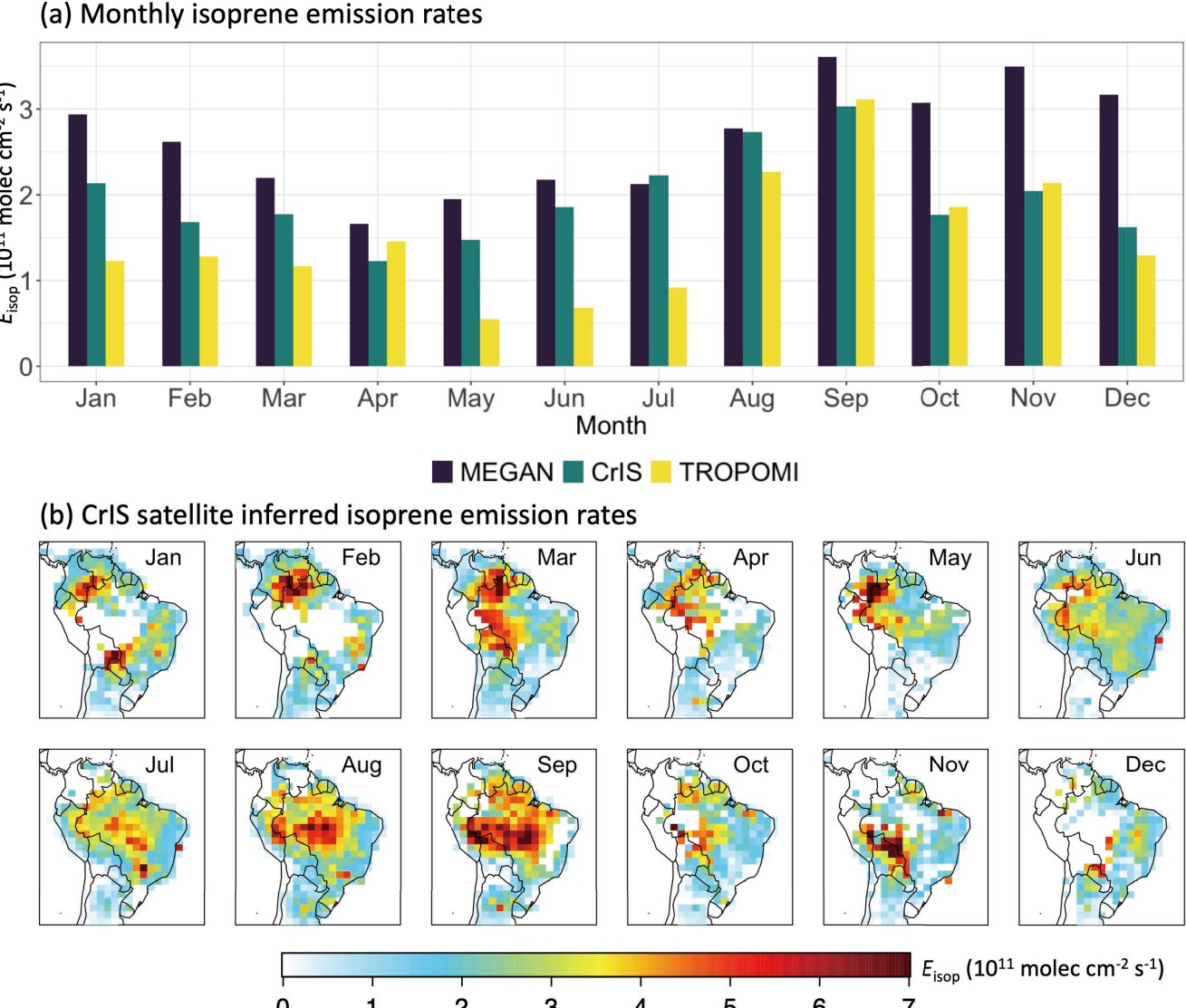

Figure 2: (a) Monthly mean MEGAN and satellite derived isoprene emission rates ($10^{11}$ molec cm$^{-2}$ s$^{-1}$) from MEGAN and as derived from CrIS isoprene and TROPOMI HCHO observations across tropical South America for 2019. (b) Monthly spatial distribution of the CrIS derived isoprene emission rates over tropical South America for 2019.

## 3.2 Evaluation of CrIS inferred isoprene emission

### Evaluation with ATTO data

We conducted nested model simulations at a horizontal resolution of 0.25° × 0.3125°, driven by MEGAN and our CrIS derived isoprene emission estimates and compared the resulting hourly isoprene mixing ratios sampled at nearest grid boxes (Fig. 3a) to the ATTO tower. We acknowledge that the CrIS $E_{isop}$ inferred at a horizontal resolution of 2° × 2.5° (Fig. 1), as described above, can only represent the mean isoprene emissions over that area. Figure 3a shows annual mean values for the enhanced vegetation index (EVI) from the MODIS instrument, which provide information about the greenness of vegetation.

The highest values of EVI over the Amazon Basin are associated with tropical rainforests that have a comparatively small seasonal variation. Measurements collected at the ATTO site should by design be representative of the surrounding rainforest within Amazon basin. As such, we assume that model isoprene mixing ratios over grid cells adjacent to ATTO, dominated by upland tropical rainforest with similar biome types, are comparable to the monthly isoprene concentrations observed at ATTO. We compare the model predicted isoprene mixing ratios averaged over the grid where the site is located

together with all the adjacent grid boxes (9 grid boxes in total from nested simulations).

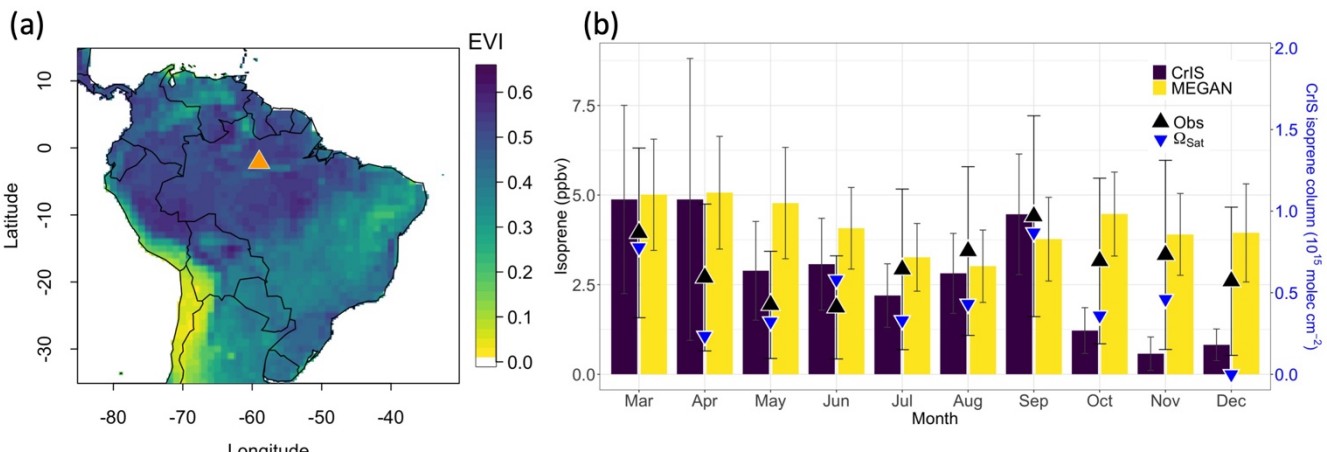

**Figure 3: (a) Annual mean values of enhanced vegetation index (EVI) from MODIS. The ATTO site is marked by an orange triangle. (b) Model and observed monthly mean isoprene mixing ratios (ppbv) at ATTO site during March-December 2019. Model values are driven by MEGAN and by values determined by the CrIS satellite data. Vertical lines denote the standard deviations of**

**the monthly means. Blue triangles denote CrIS isoprene columns at the ATTO site.**

Figure 3b shows the comparison of observed monthly mean isoprene mixing ratios at ATTO and the GEOS-Chem model during March to December 2019. The monthly satellite retrieved isoprene columns generally follow the observed monthly variations at ATTO. GEOS-Chem (MEGAN) reproduces the monthly mean ATTO data, with an annual mean isoprene mole fraction of 4.1 ± 1.3 ppbv compared with the observed annual mean value of 3.0 ± 2.2 ppbv. June and November are

transitioning months between wet (December-May) and dry season (July-October). Here we classify June as wet month and

November as dry month based on mean monthly root soil moisture at ATTO in 2019. The model (MEGAN) overestimates ATTO data by 77% during the wet months (March-June, December), $4.6 \pm 1.4$ ppbv versus $2.6 \pm 1.9$ ppbv, but is much closer during the dry months (July-November), $3.7 \pm 1.1$ ppbv versus $3.5 \pm 2.5$ ppbv. Isoprene emission estimates inferred from CrIS result in an annual mean of $2.8 \pm 1.4$ ppbv, with $2.3 \pm 1.0$ ppbv and $3.3 \pm 1.9$ ppbv during the dry (July-November) and wet (March-June, December) months, respectively. The top-down isoprene emissions underestimate the observed values from October to December partly because of low satellite observed isoprene columns as shown in Fig. 2b and because some of the grid boxes near the observational tower are set to zero emission rates where the regression relationship is not significant (p-value > 0.05), which may lower the mean simulated isoprene mole fractions at ATTO. Despite the discrepancies between model and site observations, isoprene emission estimates using CrIS isoprene retrievals can generally reproduce the magnitudes of isoprene mole fractions for most months and can better capture months with peak isoprene concentrations (March and September) compared with model using MEGAN.

We use isoprene flux measurements from the Amazon basin collected in previous years to extend our model evaluation and report the mean statistics for each site, following a previous study (Barkley et al., 2008) (see Table S1 in Supplement). Different sampling methods (e.g., technical approach, sampling height, sampling hours) can affect the magnitude of measured isoprene fluxes, and that the model isoprene fluxes are for 2019 and not from the same year as the flux measurements. We compare model isoprene fluxes during the same month and daytime hours for each flux measurement collected at different observational sites. The mean observed isoprene flux during dry months is about 3 mg $m^{-2}$ $h^{-1}$, which is generally higher than that during wet months (~1 mg $m^{-2}$ $h^{-1}$). MEGAN has higher isoprene fluxes for both seasons with 4.2 and 2.9 mg $m^{-2}$ $h^{-1}$ for dry and wet season, respectively. Satellite based isoprene flux estimates generally better reproduce the magnitudes of observed seasonal isoprene fluxes, with about 3 mg $m^{-2}$ $h^{-1}$ and 1.7 mg $m^{-2}$ $h^{-1}$ during dry and wet months, respectively. Previous studies have found that MEGAN typically overestimates isoprene fluxes over the Amazon (Bauwens et al., 2016; Gomes Alves et al., 2023). We find that CrIS-based isoprene flux estimates can potentially reduce the positive model biases in the tropical rainforest regions.

**Evaluation using TROPOMI HCHO data**

We evaluate the CrIS inferred isoprene emission rates by comparing model $\Omega_{HCHO}$ with TROPOMI $\Omega_{HCHO}$. GEOS-Chem simulated $\Omega_{HCHO}$ depends largely on isoprene emissions over the studied region, so that any reduced biases in model HCHO can be attributed to the CrIS isoprene emission estimates.

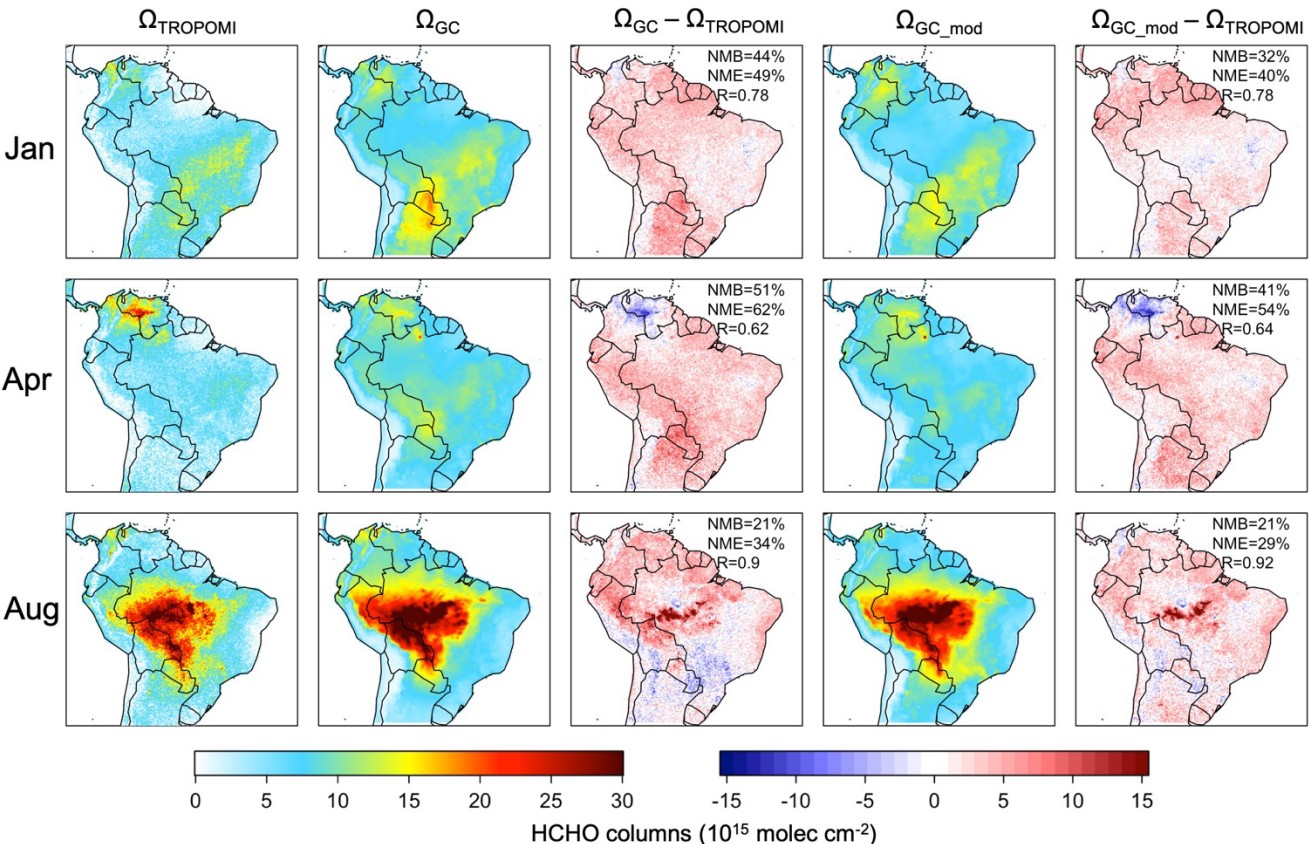

**Figure 4: TROPOMI (first column) and GEOS-Chem monthly mean HCHO columns ($10^{15}$ molec cm$^{-2}$) for Jan, April, and Aug in 2019. The GEOS-Chem model columns driven by MEGAN (second column) and by CrIS derived isoprene emission estimates (fourth column). Difference between TROPOMI and model values are shown in third and fifth columns. Shown inset of panels in the third and fifth columns are the normalized mean biases (NMB), normalized mean error (NME), and the Pearson correlation coefficients (R).**

Figure 4 shows a monthly comparison between TROPOMI HCHO columns and GEOS-Chem driven by MEGAN and satellite-based CrIS derived isoprene emissions. The model generally captures the spatial distribution of monthly HCHO columns (R = 0.61 – 0.92).  The model has a positive bias over most forested regions throughout the year, with a negative bias found over the tropical grasslands of the Colombia-Venezuelan plains to the north of the Amazon basin during March and April, as well as over the cropland to the southeast of the basin during the dry season. The CrIS derived isoprene

emission inventory reduces the annual normalised mean error (NME) from 50% to 43% and reduces the NME from 58% to 47% during the wet season (December – May) and from 36% to 33% for the dry season (July – October). Over the Amazonian region (50~75ºW, 15ºS~ 5ºN), NME is reduced from 54% to 45% annually, and from 37% to 31 during dry season, from 65% to 53% during wet season.  Table 1 shows the monthly comparison statistics between TROPOMI and

GEOS-Chem HCHO columns. We find an overall reduction in model biases, but the spatial correlation between GEOS-
Chem and TROPOMI for most months is not improved significant by using CrIS-inferred isoprene emissions. The spatial
distribution of HCHO is also strongly affected by non-biogenic sources such as wildfires and anthropogenic emissions.
Improvement in the description of the biogenic source alone does not significantly improve the overall biases. Because we
assume zero isoprene emissions where the satellite data cannot be used to derive emission rates, this approach can potentially
increase the model bias.


**Table 1: Monthly normalized mean biases (NMB), normalized mean error (NME), and the Pearson correlation coefficients (R) between GEOS-Chem simulated HCHO columns and TROPOMI HCHO columns for year 2019. For all the correlation coefficients here p-value < 0.05.**

| Isoprene emission input | | Jan | Feb | Mar | Apr | May | Jun | Jul | Aug | Sep | Oct | Nov | Dec |
|---|---|---|---|---|---|---|---|---|---|---|---|---|---|
| MEGAN 2.1 | NMB (%) | 44 | 47 | 27 | 51 | 73 | 59 | 45 | 21 | 11 | 21 | 26 | 58 |
| | NME (%) | 49 | 54 | 41 | 62 | 80 | 69 | 57 | 34 | 24 | 30 | 35 | 62 |
| | R | 0.78 | 0.71 | 0.72 | 0.62 | 0.61 | 0.65 | 0.71 | 0.90 | 0.92 | 0.82 | 0.72 | 0.66 |
| CrIS inferred | NMB (%) | 32 | 30 | 19 | 41 | 65 | 65 | 54 | 21 | 3 | –4 | 11 | 25 |
| | NME (%) | 40 | 41 | 35 | 54 | 74 | 72 | 60 | 29 | 21 | 24 | 30 | 38 |
| | R | 0.78 | 0.71 | 0.75 | 0.64 | 0.54 | 0.69 | 0.78 | 0.92 | 0.91 | 0.82 | 0.69 | 0.72 |

### 3.3 Sensitivity of isoprene emission estimates to assumed $NO_x$ emissions

Previous studies have found large scale $NO_x$ biases, likely due to underestimated soil $NO_x$ emissions, over Amazonia (Liu et al., 2016; Wells et al., 2020). $NO_x$ plays an important role in the oxidation of isoprene and thus isoprene lifetime (Atkinson, 2000; Barket et al., 2004; Lelieveld et al., 2008). The assumed model chemistry of isoprene underpins the isoprene emission estimates determined by CrIS data. A shorter isoprene lifetime will result in lower isoprene columns and consequently a
smaller slope for $\Omega_{GC} = SE_{isop,MEGAN} + B$, and a higher CrIS-based isoprene emission estimate ($E_{isop,sat} = (\Omega_{sat} - B)/S$). To examine the uncertainties from model biases in $NO_x$ emissions, we present isoprene emission estimates corresponding to a series of sensitivity tests and scale the $NO_x$ emissions by 0.25, 0.5, 0.75, 1.25, 1.5, 2, and 10. All sensitivity cases use a resolution of 2° × 2.5°. Other settings are the same as described in Sect. 2.1.

We first compare monthly GEOS-Chem simulated $NO_2$ columns with TROPOMI retrievals and find similar negative biases in model $NO_2$ (see Fig. S2 in Supplement). The comparison between GEOS-Chem and TROPOMI reflect the model bias at the satellite overpassing time. We find GEOS-Chem underestimated $NO_2$ (NMB = –16~–28%) over the Amazonian region

(50~75ºW, 15ºS~ 5ºN) during wet season (December ~ May) compared with TROPOMI. However, we find that GEOS-Chem overestimates $NO_2$ (NMB = 21~77%) during the dry season (July ~ October) in 2019 over the southern parts of Amazon, primarily due to fires. Table 2 shows the percentage changes in GEOS-Chem monthly mean $NO_2$ columns ($\Delta NO_2$) over tropical South America from our sensitivity tests compared with our base case during wet and dry months. $\Delta NO_2$ over the Amazon is shown in Table S2. We find that near to source regions, $NO_x$ emissions mostly affect the lower troposphere, as expected. GEOS-Chem $\Delta NO_2$ is typically lower during wet months due to stronger convections and to higher loss rates during wet season, particularly from deposition and atmospheric chemistry.

**Table 2: Mean relative changes (%) in simulated $NO_2$ columns under different $NO_x$ emission levels compared with the default case (EmisScale_NO = 1) during wet and dry season.**

| | | EmisScale_NO | | | | | | | |
|---|---|---|---|---|---|---|---|---|---|
| | | 0.25 | 0.5 | 0.75 | 1.25 | 1.5 | 1.75 | 2 | 10 |
| **$\Delta NO_2$** | Wet | −57% | −37% | −18% | +18% | +35% | +51% | +68% | +678% |
| | Dry | −68% | −44% | −22% | +21% | +42% | +63% | +84% | +1132% |

We derive CrIS based isoprene emission estimates for all sensitivity cases, using the method described in Sect. 2.5, to examine their sensitivity to potential biases in $NO_x$ emissions. We find that monthly spatial distributions of $E_{isop}$ generally do not vary significantly with different scaling factors (See Fig. S3 in Supplement). Table 3 summarises the relative changes in CrIS derived emission estimates ($\Delta E_{isop}$) for wet and dry seasons, under different $NO_x$ emission levels. We include the dry-to-wet and wet-to-dry months to calculate the annual mean $\Delta E_{isop}$. Satellite predicted values of $E_{isop}$ increase (decrease) with higher (lower) $NO_x$ emissions. Figure 5b summarises the monthly $E_{isop}$ and the corresponding isoprene lifetimes from all model grids over tropical South America, showing that lower $NO_x$ emissions (EmisScale_NO < 1) generally have longer isoprene lifetimes and lower predicted isoprene emission estimates. Underestimated $NO_x$ emissions over the Amazon can lead to higher isoprene lifetime and thus lower predicted isoprene emission rates.

**Table 3: Mean relative changes (%) in monthly CrIS derived isoprene emission rates under different $NO_x$ emission levels compared with the default case (EmisScale_NO = 1) during wet and dry season.**

| | | EmisScale_NO | | | | | | | |
|---|---|---|---|---|---|---|---|---|---|
| | | 0.25 | 0.5 | 0.75 | 1.25 | 1.5 | 1.75 | 2 | 10 |
| **$\Delta E_{isop}$** | Wet | −13% | −7% | −0.1% | +8% | +9% | +17% | +21% | +96% |
| | Dry | −25% | −15% | −6% | +7% | +13% | +20% | +26% | +146% |
| | Annual | −19% | −10% | −1% | +10% | +13% | +21% | +26% | +118% |

To compensate for the $NO_2$ column model negative bias (NMB = –16~–28%) over the Amazon during wet season and the $NO_2$ column model positive bias (NMB = 21~77%) during the dry season over the southern Amazon, we use scale factors of 1.25 ($\Delta NO_2$ = +18% for wet season) and 0.25 ($\Delta NO_2$ = –68% for dry season) to estimate CrIS based isoprene emission

estimates. The predicted satellite based emission rates would be increased by ~8% during the wet months, and be reduced by ~25% during dry months, accordingly (Table 3). Figure 5a shows the monthly mean CrIS isoprene emission estimates corresponding to our sensitivity cases over the Amazon. We find that the seasonal variation of the isoprene emissions estimates that correspond to the $NO_x$ sensitivity experiments follow our base case (Fig. 3). Considering the potential model biases in $NO_x$ emissions, satellite based isoprene emission rates can vary by about ± 20% annually.

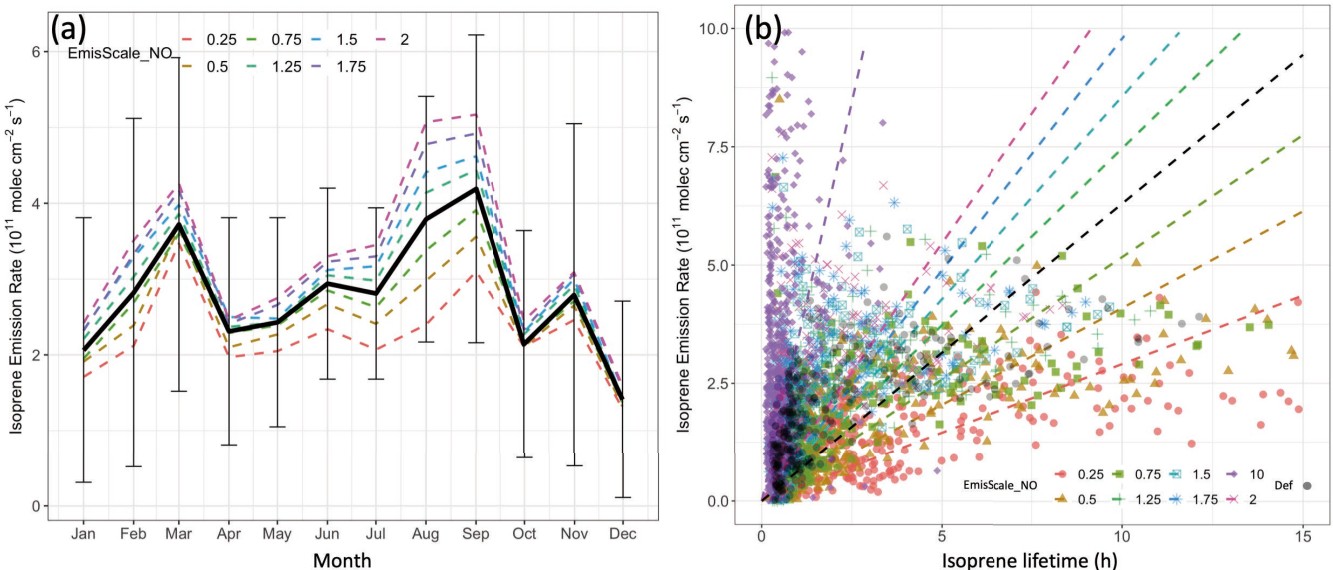


**Figure 5: (a) Monthly mean CrIS based isoprene emission rates rates ($E_{isop}$, $10^{11}$ molec cm$^{-2}$ s$^{-1}$) over the Amazonian region under different $NO_x$ emission levels. Black line indicates monthly mean with standard deviations of the default case (EmisScale_NO = 1). (b) Annual mean CrIS derived isoprene emission rates vs. isoprene midday lifetime under different $NO_2$ emission levels over tropical South America. Default CrIS isoprene emission rates is shown in black.**


Figure 6 shows monthly isoprene emission rates inferred from CrIS isoprene column data for which grid-dependent $NO_x$ emissions are scaled using TROPOMI tropospheric $NO_2$ columns. These grid-dependent scaling factors are determined by monthly $NO_2$ column differences between GEOS-Chem and TROPOMI (Fig. S2). For example, for a model grid where the monthly TROPOMI $NO_2$ column is 75% lower than the corresponding GEOS-Chem value, we scale the model $NO_x$

emissions by a factor of 0.25 (EmisScale_NO = 0.25). Using this approach, we account for the spatial distribution of model biases in $NO_x$ emissions. As a caveat, scaling the $NO_x$ emissions in the model in this way does not necessarily reflect the real emission biases. For example, the influence of convection and advection are not considered in the distribution of atmospheric $NO_x$. Moreover, these scaling factors are calculated at the satellite overpass time and consequently do not

represent any time-dependence in model bias. We examine the monthly simulated isoprene mole fractions with scaled $NO_x$

emissions at the ATTO site. The scaling factor is 1.25 for March to May, 0.75 for July to September, and 1 for June and October to December based on the monthly differences between model and satellite $NO_2$ columns at ATTO. The resulting model bias is reduced for wet and dry months. For wet months (March~May), the mean model isoprene mole fraction is reduced from 4.2 to 3.8 ppbv, corresponding to higher $NO_x$ emission levels, compared with the observed value of 2.9 ppbv. For dry months (July~September), the mean model isoprene mole fraction is increased from 3.2 to 3.6 ppbv, corresponding

to lower $NO_x$ emission levels, closer to the observed value of 3.6 ppbv.

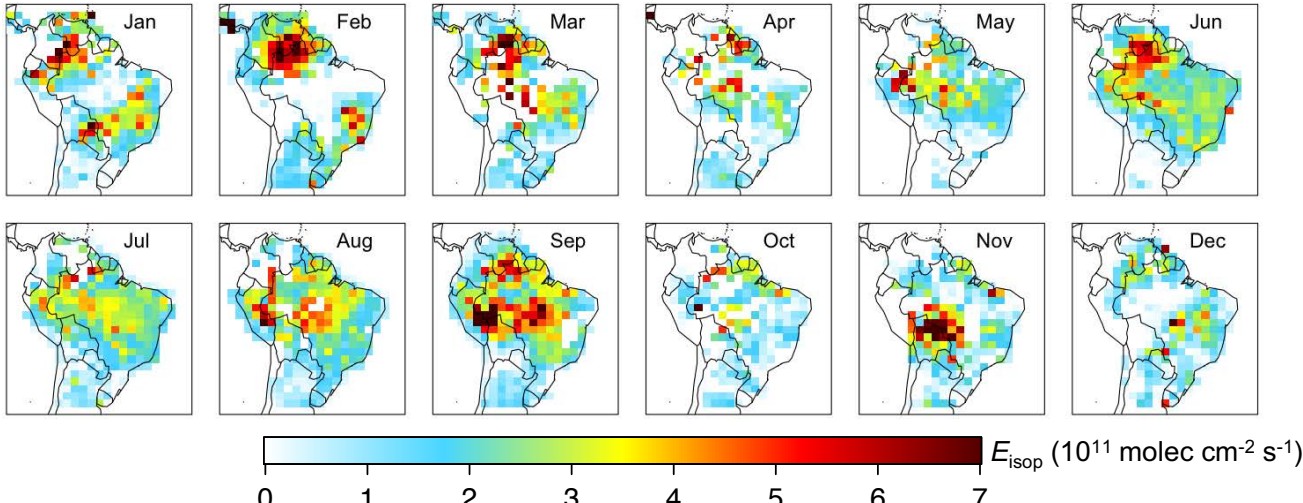

**Figure 6: Monthly CrIS derived isoprene emission rates over tropical South America for 2019 using grid-dependent scaled $NO_x$ emissions.**

## 4 Concluding Remarks

Using the GEOS-Chem atmospheric chemistry transport model, we derived top-down isoprene emissions over tropical South America for 2019 using isoprene columns retrieved from CrIS on the NOAA-20 satellite. We found that isoprene emission estimates inferred from CrIS data result in very different spatial and seasonal distributions of isoprene columns over tropical South America than when we use the MEGAN isoprene emission inventory.

We evaluated our CrIS derived isoprene emissions by comparing the corresponding isoprene concentrations with observations collected at the Amazon Tall Tower Observatory, March-December 2019, and found the CrIS derived isoprene emissions reproduce the magnitude of the seasonal cycle better than MEGAN, with smaller monthly biases. The CrIS derived isoprene emission inventory was evaluated by comparing modelled HCHO distributions based on itself against TROPOMI HCHO. We found that this isoprene emission inventory reduced the annual normalised mean error from 50% to

43%, relative to MEGAN, and reduced the NME from 58% to 47% during the wet season (December – May) and from 36% to 33% for the dry season (July – October). We find that the satellite-derived isoprene emission estimates improve the model ability describe monthly variations, with individual monthly values varying by about ± 20% with model $NO_x$ emission biases. We find that accounting for model biases in $NO_x$ emissions using satellite retrieved $NO_2$ columns can potentially improve the satellite derived isoprene emissions.

More accurate estimates of isoprene are of great importance for understanding the relative contribution of anthropogenic and biogenic sources to the formation of ozone and secondary organic aerosol in the upper troposphere (Palmer et al., 2022; Curtius et al., 2024; Shen et al., 2024). Human induced land use and land cover changes have been found to strongly influence isoprene emissions during recent decades compared with those induced by climate change (Chen et al., 2018). Satellite retrievals of isoprene columns, interpreted using state-of-the-art atmospheric chemistry transport models, can help understand some of the impacts on atmospheric composition from, for example, continuing deforestation, widespread drought, and heatwaves. Recent work has shown that these data can also track changes in atmospheric oxidation over forested regions (Shutter et al., 2024). Tracking changes in isoprene over tropical rainforests, in the context of wider land surface quantities, provides a way to check on the health of these remote ecosystems. The Amazon basin has suffered from severe droughts in recent years, associated with deforestation and changes in climate (Bottino et al., 2024; Espinoza et al., 2024). The El Niño-Southern Oscillation has also contributed to droughts in Amazonia, and is predicted to induce more extreme heatwaves and floods over this region in the future (Marengo and Espinoza, 2016). At a time when we are witnessing such extensive and widespread environmental change, particularly across the tropics, it is essential we sustain these important environment datasets (Millet et al., 2024) because they may be one of the first harbingers of an emerging feedback (Spracklen and Coelho, 2023).

**Data availability**. GEOS-Chem model code and input data are available from the GEOS-Chem website (www.geoschem.org). TROPOMI HCHO and $NO_2$ data used in this study can be found at https://doi.org/10.5270/S5P-s4ljg54 (Koninklijk Nederlands Meteorologisch Instituut (KNMI), 2018) and https://doi.org/10.5270/S5P-tjlxfd2 (German Aerospace Center (DLR), 2019) respectively. The CrIS isoprene data are to be archived on the Centre for Environmental Data Analysis (CEDA) archive (https://archive.ceda.ac.uk/).

**Author contributions.** S. S. led the experimental design and the associated model runs and data analysis. P. I. P. helped to design the study and wrote the manuscript with S. S.. All authors helped to revise the manuscript. R. S., B. J. K., and L.V. provided the CrIS isoprene data and the support necessary to apply these data to study isoprene emissions. A. E., A. R., E. Y. P. and J. W. contributed isoprene measurements at ATTO tower.

**Competing interests.** At least one of the (co-)authors is a member of the editorial board of Atmospheric Chemistry and Physics. The authors have no other competing interests to declare.

**Acknowledgements.** This research was supported by the UK National Centre for Earth Observation (NCEO) funded by the Natural Environment Research Council (NE/R016518/1).

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
