# Peer review of "Seasonal isoprene emission estimates over tropical South America inferred from satellite observations of isoprene"

_EGUsphere, 2025_

## Referee Comment (RC2)

Sun et al. (2025) present a comparison of bottom-up and top-down isoprene emission estimates over tropical South America for 2019. The work is the first to use the CrIS retreivals from RAL's MLS scheme for this purpose. The model performance using updated emissions is compared to remotely sensed HCHO and in-situ isoprene measurements. The authors discuss potential drivers of biases in bottom-up emission estimates, and potential impacts on modeled ozone production sensitivities in the region.

Overall, the work is on an important topic and suitable for the audience of ACP. Additional details are needed to verify methodological soundness. Specific comments are given below.

**Major comments**

(1) More details are needed to confirm the maturity of the CrIS isoprene product used in the top-down analysis. The manuscript refers the reader to a Palmer et al (2022), and in the supplement of that paper, there is a discussion of the retrieval method; however, the uncertainties in the retrieval are not sufficiently detailed for the foundation of a top-down emissions study. For example, this manuscript states that "IMS column averages then to be lower than those derived from surface-based observations". It seems almost to be expected that IMS-based emissions would be lower than those from bottom-up emissions inventories. However, it is not clear how large these biases are. To address this concern, I suggest the authors (1) provide a quantitative assessment of measurement uncertainty (2) providing a comparison with other CrIS isoprene products, which may have more validation studies.

(2) It seems flux measurements are a more direct method of evaluating the top-down emissions estimates rather than relying on concentration measurements. Flux measurements have been reported previously for the ATTO tower (e.g., Langford et al., 2022). It may be that there were not flux measurements available in during the time period of this study- I am not sure. However, to whatever extent possible, the authors are encouraged to leverage observation-based assessments of isoprene fluxes to contextualize their work.

(3) The GEOS-Chem model simulation requires further evaluation. Doe the model systematically misrepresent NOx concentrations and/or PBL height? If so, how sensitive are the results to any biases in these parameters? Are there other measurements (e.g., isoprene oxidation products, ozone?) that could be used to assess model performance?

(4) The manuscript compares NME between TROPOMI and GEOS-Chem as a method of assessing the improvement offered by top-down emissions. Because difference in spatial patterns are also highlighted as a result, it follows that spatial correlation between GEOS-Chem and TROPOMI in the top-down and bottom-up scenarios should also be addressed within the main body of the manuscript. Table S1 in the supplement suggest the bottom-up inventory may produce a similar, and even sometimes better, correlation. How do the authors interpret this result?

(5) I do not find the HCHO:NO2 work to contribute meaningfully to the aims of this paper. There is a cursory acknowledgment in the problems associated with threshold-based analysis (line 340), but those caveats are not considered addressed or considered. The use of fixed numerical values (line 365) from a paper centered on China should either be removed, or rationalized more clearly.

**Minor and technical comments**

Line 33: A lifetime of ~1 hr is given for isoprene. According to the GEOS-Chem model, what is the range in mid-day isoprene lifetime in gridcells over tropical South America?

Line 97: More detail is required on the "standard" HEMCO configuration, as the manuscript currently presumes the reader knows what is included, and "standard" could change. For example, is CO2 inhibition or drought response included? Where it the LAI data from? Is this offline emissions using GEOS-FP meteorology?

Line 139. "Although the a priori constraint on the retrieval is weak, this is also accounted for": I am not sure what "this" refers to, what "accounted for" means, and how that is accomplished. Please clarify.

Line 163: For what regions was this bias-correction formula derived for? How do we know they are applicable here?

Line 172: Isoprene measurements from three measurement heights are used. How are these matched to GEOS-Chem levels? How is the data aggregated into the results shown in figure 3? Is there a sharp spatial gradient seen?

Line 174: Notation on the Ringsdorf citation needs to be corrected.

Section 2.5: What regression method was used? Does it incorporate error in the x and y axis? The paragraph starting at line 194 and the first paragraph were somewhat redundant. Combining these could help with readability.

Line 326: The third sentence of this paragraph is a fragment.

---

## Author Comment (AC1)

**Response to referee comments on "Seasonal isoprene emission estimates over tropical South America inferred from satellite observations of isoprene"**

Sun et al.

We thank both referees for their insightful and helpful comments. Below we have responded to all individual comments. Our responses are written in blue. Text from the manuscript is shown *italicized* with new text added to the manuscript *italicized and underlined*.

**Referee #1:**

This study employs modeled isoprene column:emission rations from GEOS-Chem to derive monthly isoprene emissions for 2019 over tropical South America based on the RAL IMS CrIS isoprene column retrieval. The authors evaluate their results using in situ isoprene concentrations at the ATTO site in Brazil as well as TROPOMI HCHO columns. The CrIS-based isoprene emissions resulted in modest improvements in model HCHO and isoprene with respect to MEGAN, with a small impact on model ozone over the region.

This study presents a novel use of a new satellite product and is well within the scope of ACP. However, I think the manuscript needs significant work before it can be considered for publication. My main concern is that it fails to consider the potential impact of documented low model NOx biases in the region, and how the resulting high isoprene lifetimes will bias the emissions derived from CrIS based on the model isoprene column:emission slopes. The authors are strongly encouraged to first consider adjusting the model NOx emissions based on a comparison to TROPOMI NO2, or at least perform a sensitivity study in which NOx emissions are adjusted to quantify how much it impacts their findings.

We thank the reviewer for the constructive suggestion to consider the model biases in NOx. To help address this concern, we have added a comparison of model $NO_2$ and TROPOMI $NO_2$ across our study region and reported sensitivity experiments in which we adjust NOx emissions using scale factors in GEOS-Chem. We have also analysed the impacts of changes in NOx emissions to the lifetime of isoprene and how this affects satellite derived isoprene emission rates. The revised manuscript now includes discussion about potential NOx biases and how this affects the satellite based isoprene emission rates in our study.

**TROPOMI $NO_2$ vs GEOS-Chem $NO_2$**

As the reviewer pointed out, previous studies have found large scale NOx biases likely due to underestimated soil NOx emissions over Amazonia (Liu et al., 2016; Wells et al., 2020). We found similar negative biases in simulated $NO_2$ as shown in Fig. 1, reproduced below. GEOS-Chem underestimated $NO_2$ (NMB = –16~–28%) over the Amazonian region (50~75ºW, 15ºS~ 5ºN) during wet season (December ~ May) compared with TROPOMI. However, the

model overestimate $NO_2$ (NMB = 21~77%) caused by wildfires during the dry season (July ~ October) in 2019 over the southern parts of Amazon.

[Figure]

**Figure 1. Comparison between (a) TROPOMI and (b) GEOS-Chem $NO_2$ columns. (c) the difference between model and satellite $NO_2$ columns.**

**Sensitivity analysis**

To investigate the uncertainties from model biases in NOx emissions, we carried out a series of sensitivity tests and scaled the NOx emissions with scale factors of 0.25, 0.5, 0.75, 1.25, 1.5, 2, and 10 to represent different levels of NOx emission levels. For example, a scale factor of 1.5 would scale all NO emissions up by 50%. All sensitivity cases use a resolution of 2º × 2.5º. Other settings are the same as described in Sect. 2.1.

The relative changes in simulated mean $NO_2$ columns $\Delta NO_2$ (%) over tropical South America in sensitivity tests compared with the default case during wet and dry months are shown in Table 1 below. The comparison between model and TROPOMI and the biases shown in Fig. 1 only reflect the potential model biases at the satellite overpassing time. The modelled $NO_2$ mixing ratios within mid to lower troposphere change with NOx emissions proportionally, yet they are less affected at higher levels.

**Table 1: Relative changes (%) in simulated $NO_2$ columns under different NOx emission levels compared with default case (EmisScale_NO = 1).**

| | | EmisScale_NO | | | | | | | |
|---|---|---|---|---|---|---|---|---|---|
| | | 0.25 | 0.5 | 0.75 | 1.25 | 1.5 | 1.75 | 2 | 10 |
| $\Delta NO_2$ | Wet | −57% | −37% | −18% | +18% | +35% | +51% | +68% | +678% |
| | Dry | −68% | −44% | −22% | +21% | +42% | +63% | +84% | +1132% |

The dominant loss of isoprene is the reaction with OH during daytime. Here we compare monthly mean isoprene lifetime ($\tau_{isop}$, hr) against OH oxidation ($\tau_{isop} = (k_{isop}[OH])^{-1}$) within the boundary layer for our sensitivity simulations. $k_{isop}$ is the rate constant for the reaction of isoprene with OH. $k_{isop} = 2.69 \times 10^{-11} \exp(390/T)$, where T is temperature (K). Figure 2a shows monthly midday (13:00 LT) isoprene lifetime over the studied region from sensitivity tests, while Fig. 2b shows monthly distributions of $\tau_{isop}$ with default NOx emissions. Model simulates longer $\tau_{isop}$ under lower NOx emissions and vice versa. The Amazon region has the highest $\tau_{isop}$ throughout the year. Underestimated NOx emissions over the Amazon can lead to higher $\tau_{isop}$ during some wet months (Mar–May).

[Figure]

**Figure 2: (a) Simulated monthly mean midday isoprene lifetime (hr) over Tropical South America with different NO$_x$ emission levels. Error bars indicate standard deviation over the studied region. (b) Monthly distribution of mean isoprene lifetime (hr) with default NOx emissions (EmisScale_NO = 1).**

We derived CrIS based isoprene emission rates for all sensitivity cases, using the method described in Sect. 2.5, to show how potential biases in NOx emissions affect the CrIS based isoprene emission rates (Fig. 3 below). The relative changes in CrIS derived emission rates ($\Delta E_{isop}$) under different NOx emission levels are summarised in Table 2 below.

[Figure]

**Figure 3. Monthly CrIS derived isoprene emission rates ($E_{isop}$, $10^{11}$ molec cm$^{-2}$ s$^{-1}$) with different $NO_2$ emission level (scale factor = 0.25, 1, 2).**

To compensate for the $NO_2$ column model negative bias (NMB = –16~–28%) over the Amazon region during wet season (December ~ May) and the $NO_2$ column model positive bias (NMB = 21~77%) during the dry season (July ~ October) over the southern parts of Amazon (Fig. 1), we use scale factors of 1.25 ($\Delta NO_2$ = +18% for wet months) and 0.25 ($\Delta NO_2$ = –68% for dry months) to estimate CrIS based emission rates. Table 2 below shows the predicted satellite based emission rates would be increased by ~8% during the wet months, and be reduced by ~25% during dry months accordingly. Monthly mean satellite based isoprene emission rates derived from sensitivity cases over the Amazon is shown in Fig. 4 below. Model with higher emission scale factors predicts higher isoprene emission rates. The month-to-month variations (Mar~Dec) of CrIS based emission rates also follow the observed isoprene mixing ratios shown in Fig. 3 in the main text. Considering the potential model biases in NOx emissions, satellite based isoprene emission rates can vary by about ± 20% annually.

**Table 2: Relative changes (%) in monthly CrIS derived isoprene emission rates under different NOx emission levels compared with default case (EmisScale_NO = 1).**

| | | EmisScale_NO | | | | | | | |
| --- | --- | --- | --- | --- | --- | --- | --- | --- | --- |
| | | 0.25 | 0.5 | 0.75 | 1.25 | 1.5 | 1.75 | 2 | 10 |
| $\Delta E_{isop}$ | Wet | −13% | −7% | −0.1% | +8% | +9% | +17% | +21% | +96% |
| | Dry | −25% | −15% | −6% | +7% | +13% | +20% | +26% | +146% |
| | Annual | −19% | −10% | −1% | +10% | +13% | +21% | +26% | +118% |

[Figure]

**Figure 4: Monthly mean CrIS based isoprene emission rates over the Amazonian region under different NO$_x$ emission levels. Black line indicates monthly mean with standard deviations of the default case (EmisScale_NO = 1).**

Figure 5 summarises the relationships between isoprene emission rates inferred from CrIS data and model isoprene lifetimes under different NOx emission levels. The isoprene lifetime is related to the simulated isoprene columns used in this study to derive satellite based isoprene emissions. Shorter isoprene lifetime and corresponding lower isoprene columns result in smaller slopes in $\Omega_{GC} = SE_{isop,MEGAN} + B$, and the corresponding CrIS based emission rates would be higher ($E_{isop,sat} = (\Omega_{sat} - B)/S$). Simulations with reduced NOx emissions (EmisScale_NO < 1) have longer isoprene lifetimes and lower predicted isoprene emissions in general.

[Figure]

**Figure 5: Isoprene emission rates vs. isoprene lifetime in sensitivity cases. Annual mean CrIS derived isoprene emission rates ($E_{isop}$, $10^{11}$ molec cm$^{-2}$ s$^{-1}$) under different $NO_2$ emission levels over tropical South America. Default CrIS isoprene emission rates (EmisScale_NO = 1) is shown in black.**

I also think that this study would be improved by first including an evaluation of the RAL IMS isoprene product, as it is a new retrieval for which no validation has yet been published. The text could also use some clarification throughout; see below for specific recommendations.

Thanks for the comment. The submitted manuscript already includes an indirect evaluation of the RAL IMS isoprene product, as described below, but to make this clearer we have now added a section of data evaluation. All the comparisons described take into account the instrument averaging kernel, without which any comparison with independent data has limited worth.

Evaluating satellite observations of isoprene column data is difficult because the vast majority of the data that are collected is typically on the ground or on tall towers. In this case, we can use the GEOS-Chem as an intermediary to relate surface data and isoprene column data. This approach has been already used with the ATTO data collected during our study year in Fig. 3 in the main text. We showed that the isoprene emission estimates inferred from the satellite

data are consistent with the surface data. The seasonal estimates of isoprene mixing ratios from satellite based emissions are also more consistent with observations than MEGAN inventory.

We extend this approach to include direct isoprene flux measurements, collected in previous years and report the mean statistics as shown in Table 3 below, following work reported by Barkley et al. (2008). Note that different sampling methods (e.g., technical approach, sampling height, sampling hours) can affect the magnitude of measured isoprene fluxes, and that the model isoprene fluxes are not for the same year as the measurements. Here we compare model values during the same month and daytime hours for each flux measurement collected at different observational sites. The observed isoprene flux is generally higher during the dry season than the wet season. Satellite based isoprene flux estimates generally reproduce the magnitudes of observed seasonal isoprene fluxes. Previously studies have found that MEGAN typically overestimates isoprene fluxes over the Amazon (Bauwens et al., 2016; Gomes Alves et al., 2023). We find that CrIS-based isoprene flux estimates can potentially reduce the positive model biases in the tropical rainforest regions.

**Table 3: Direct isoprene flux measurements and modelled isoprene flux.**

| Location | Season | Sampling period | Sampling method | Observed isoprene flux [mg m$^{-2}$ h$^{-1}$] | MEGAN isoprene flux* [mg m$^{-2}$ h$^{-1}$] | CrIS-based isoprene flux* [mg m$^{-2}$ h$^{-1}$] | References |
|---|---|---|---|---|---|---|---|
| ATTO site 150km northeast of Manaus, Brazil | Dry-to-wet | 11–21 Nov 2015 | Mean daytime (09:00–17:00 LT); Eddy covariance | 3.1 | 3.0 | 1.2 | Alves et al. (2023) |
| Cuieiras Biological Reserve (TT34-ZF2), Manaus, Brazil | Dry Wet | Jan – Dec 2013 | Mean daytime (09:00–17:00 LT); disjunct eddy covariance technique | 2.8 1.9 | 3.5 3.0 | 2.5 2.7 | Langford et al. (2022) |
| Cuieiras Biological Reserve (TT34-ZF2), Manaus, Brazil | Dry Dry-to-wet Wet | Sep–Oct 2010 Nov 2010 Dec 2010–Jan 2011 | Mean daytime (10:00–14:00 LT); PTR-MS, gradient profile and gradient flux | 1.4 1.4 0.5 | 4.8 4.5 3.7 | 3.9 1.6 1.3 | Alves et al. (2016) |
| Cuieiras Biological Reserve (TT34-ZF2), Manaus, Brazil | Dry-to-wet | Sep–Dec 2010 | Mean daytime (10:00–16:00, LT) at 35m; PTR-MS, gradient profile and gradient flux | 1.4 | 3.8 | 2.1 | Jardine et al. (2012) |
| Cuieiras Biological Reserve (C14-ZF2), Manaus, Brazil | Dry | Sep 2004 | Mean daytime (12:00–14:00, LT); PTR-MS, disjunct eddy covariance | 8.3 | 5.6 | 6.1 | Karl et al. (2007) |
| Cuieiras Biological Reserve (K34- | Dry | Jul 2001 | Mean daytime (10:00–15:00 LT); | 2.4 | 4.3 | 3.7 | Kuhn et al. (2007) |

| | | | | | | | |
|---|---|---|---|---|---|---|---|
| ZF2), Manaus, Brazil | | | GC-FID, relaxed eddy accumulation | | | | |
| Tapajós National Forest, Brazil | Wet-to-dry | Jun 2014 | Daytime; PTR-TOF-MS, eddy covariance (~65m) | 0.7 | 2.1 | 1.0 | Sarkar et al. (2020) |

**\*MEGAN isoprene flux and CrIS based isoprene flux are for year 2019.**

We also used the model as an intermediary to test the consistency between IMS data and TROPOMI formaldehyde columns. We acknowledge this test also assumes perfect knowledge of the intermediate atmospheric chemistry, but we show that isoprene emissions inferred from the IMS data are more consistent with TROPOMI than the MEGAN inventory. In the revised manuscript we have clarified this comparison as a form of evaluation of the IMS data.

A comparison between the RAL IMS data product and the independent data product developed by the University of Minnesota is less trivial because their current retrieval approach does not include an averaging kernel. So any comparison would not be easy to interpret.

**Specific comments**

Line 54-55: What is meant by "chemical networks"? Do the authors mean "chemical mechanisms" here?

Thanks for the comment. We have made the correction "*chemical mechanisms*". The confusion stems from studying planetary atmospheres where the community refer to chemicals networks while Earth-focused studies talk about chemical mechanisms.

Line 65: Since there are a few different CrIS isoprene retrievals available, it would be good to mention in the introduction which is used in this work.

Good suggestion. We have now added descriptions of CrIS isoprene retrievals and corresponding references in the introduction:

> "*Fu et al. (2019) developed the first direct isoprene retrievals using infrared radiance measurements from CrIS. The Retrieval of Organics from CrIS Radiances (ROCR) isoprene product is built upon Fu et al. (2019) and uses a machine learning algorithm to derive isoprene abundances from the CrIS hyperspectral range index (HRI) (Wells et al, 2022).*"

Line 105: What soil NOx emission scheme was used in the simulation? This is important to know given the sensitivity of the column:emission relationship to isoprene lifetime.

We use Hudman et al. (2012) soil NOx scheme. We now mention this in the revised manuscript:

> "*The offline soil $NO_x$ emissions used in this study is generated with GEOS-FP (0.25° × 0.3125°) meteorology field resolution with Hudman et al. (2012) scheme.* "

We have now added sensitivity tests and analysed how isoprene lifetime is affected by potential model biases in soil NOx emissions.

Line 107-122: Some of this information would be better suited to the introduction rather than contained in the methods, since it reflects general uncertainties in our understanding of isoprene emission processes that are not unique or specific to the MEGAN model being introduced here.

Agreed. We have modified this paragraph accordingly and moved a general description of MEGAN to the introduction.

Line 128: What do the authors mean by "replace any a priori information assumed by the retrieval" here? Don't the authors apply both the averaging kernel and a priori profile from the RAL IMS retrieval to the GEOS-Chem profile in their comparisons?

We agree this is vague.

The RAL IMS retrievals use an optimal estimation approach for which we use *a priori* isoprene reference profile (a constant mixing ratio of isoprene) to calculate the retrieved isoprene scale factor. To calculate the corresponding model isoprene profiles, we apply this scale factor together with averaging kernels. We use the following equations:

$$A_{isop} = G_{isop} K_{isop},$$

where $A_{\text{isop}}$ is the averaging kernel, $K_{\text{isop}}$ denotes the weighting function matrix elements, $G_{isop}$ is the sensitivity of the retrieved isoprene scale factor with respect to changes in the measurement vector, representing one row of the retrieval gain matrix:

$$G = \left(K^T S_y^{-1} K + S_a^{-1}\right)^{-1} K^T S_y^{-1},$$

where weighting function $K$ includes the isoprene scale factor and thus the assumed *a priori* profile. $S_y$ is the measurement error covariance matrix. $S_a$ is the a priori error covariance matrix.

$$\Omega_{isop}^{GC} = A_{isop} n_{isop}^{GC},$$

where $\Omega_{isop}^{GC}$ is the modelled isoprene column, $n_{isop}^{GC}$ is the modelled isoprene profile. A more detailed technical description can be found in the supplementary text in Palmer et al. (2022).

To address this reviewer comment, we have provided additional information in the revised manuscript.

Line 133: There is also a CrIS instrument onboard the NOAA-21 satellite, launched in 2022.

Thanks for this information. We have added this in the revised manuscript.

Line 142-143: "IMS column averages tend to be lower than those derived from surface-based observations when surface level concentrations are high…" I think the authors need to do more to put this statement into context, as it does not necessarily represent a problem and is in fact

expected behavior for IR satellite retrievals that are more sensitive to the mid-troposphere. A reader might read this and incorrectly assume the retrieval has a low bias, when there has actually been no published validation (that I know of) for the RAL IMS isoprene retrieval. The latter fact also bears mentioning here.

We have added explanation about the RAL IMS isoprene retrievals:

"… *IMS column averages tend to be lower than those derived from surface-based observations where surface level concentrations are high, as expected. The sensitivity of infrared spectra to trace gases is generally lowest near the ground because of the small temperature difference between the atmosphere and the surface, particularly at night. In this study, we use daytime satellite retrieved isoprene columns which correspond with peak isoprene emissions.*"

Line 164: In addition to the bias correction to OMI and TROPOMI, does the HCHO comparison also include an application of observation operators to the GEOS-Chem profiles?

We apply corresponding averaging kernels to the GEOS-Chem simulated HCHO profiles and all other satellite retrievals used in this study. We added clarification in the revised text:

Line 182: I assume that B and S refer to the intercept and slope of the linear regression model here, but the authors should still define these variables.

We have added the physical meaning of these two quantities in the revised manuscript.

"… *The intercept B refers to the isoprene background, while the slope S refers to the isoprene column corresponding to the isoprene emission rates.* … "

Line 183-186: As the authors note, the slope in this isoprene column:isoprene emission comparison is mainly determined by the isoprene lifetime, and the slope in the HCHO column:isoprene emission comparison is determined both by the HCHO yield and lifetime. How do the authors account for the fact that these may be biased in the model, due to biases in NOx for example? These would potentially lead to large differences in the slope that is being used to derive emissions from the satellite observations.

This is a great point raised by the reviewer. Please refer to the discussion above. We have revised the manuscript accordingly.

Figure 1: Since the GEOS-Chem output has had scene dependent averaging kernels applied, I assume the monthly mean only reflects times when the CrIS data were also available (i.e. model fields were screened when CrIS data were discarded due to cloud or other quality concerns). Is that correct?

Yes, the GEOS-Chem model fields were screened where CrIS data were absent or did not pass the quality thresholds. We have now clarified this is the manuscript.

Line 270-274: These sentences are unclear. Are the authors saying that they compare the ATTO measurements to an average of the GEOS-Chem predictions for the grid box containing the site, and all adjacent grid boxes?

We compared the nested model predictions averaged over the grid where the site is located together with all the adjacent grid boxes (9 grid boxes in total at 0.25º × 0.3125º). We have rephrased this in the revised manuscript.

Lines 309-323: I found this discussion to be too general to be very useful to the reader. Also, the parameters in Figs 3c-f will better correlate with isoprene emissions than concentrations, so if the authors want to include them it would make more sense to do so in Figure 2. Most importantly, however, this section is missing a discussion of model NOx biases (and, thus, biases in isoprene lifetime) as a possible source of discrepancies between the observed and simulation concentrations. Assuming that the Hudman et al. (2012) soil NOx emission scheme was used in this work, there has been at least one study that found it significantly underestimates NOx (by a factor of 30) in the region (Liu et al., 2016). I strongly encourage the authors to include a sensitivity study adjusting the NOx in their simulation to see how it impacts their results.

Thanks for the suggestion. We have removed the environmental parameter discussion and Figs 3c-f. We have included the sensitivity analysis to replace the discussion of HCHO:NO$_2$ ratios in the revised manuscript.

Lines 335-344: While the CrIS-derived emissions yield modest improvements in the mean bias and error of GEOS-Chem HCHO with respect to TROPOMI, the spatial correlation is often unchanged or even degraded. Do the authors have some ideas as to why this is?

As the referee pointed out: CrIS-derived emissions have unchanged correlation compared with default emissions (typo in Supplement where CrIS-inferred correlation $R$ is supposed to be 0.78 for Jan) except for degraded correlation over May (CrIS $R = 0.54$; MEGAN $R = 0.61$) and Nov (CrIS $R = 0.69$; MEGAN $R = 0.72$). The distribution of monthly HCHO is also strongly affected by non-biogenic sources such as wildfire as well as anthropogenic emissions such as transportation and industrial processes. Improvement in biogenic source alone doesn't improve the overall biases significantly. As we assume zero isoprene emissions where satellite data cannot be used to derive emission rates, this potential underestimation can also increase biases.

Lines 354-386: This section also needs to consider the potential impact of model NOx biases on the FNR and the resulting model ozone. Could the authors evaluate the model NO2 based on the TROPOMI NO2 and adjust NOx emission accordingly to see how it improves the model ozone with and without the CrIS-based isoprene emissions?

We've adopted suggestion from review #2 that the HCHO:NO$_2$ work does not contribute meaningfully to the aims of this paper and we decided to remove this section and replace with the sensitivity tests based on the analysis in our response above.

**Technical comments**

Line 18: Suggest changing "north of Amazon" to "northern Amazon" and changing "southeast of Brazil" to "southeast Brazil"

Line 31: "Influences" should be "influence"

Line 64: Insert "the" before "GEOS-Chem"

Line 132: Insert the word "launched" before "onboard"

Line 147: "am equatorial" should be "an equatorial"

Line 148: "collected" should be "collects"

Line 152: "We refer to the reader to a dedicated reported" should be edited to "We refer the reader to a dedicated report"

Line 175: Add parentheses around the year for the citation here.

Line 178: "relationships these" should be edited to "relationship of these"

Line 222: I think "NMB > 100%" should be "NMB > 0%" here? The NMB values reported earlier in the paragraph are both positive (> 0) but less than 100%.

Line 254: Insert "are" between "hotspots" and "collocated"

Line 271: Insert "the" before "ATTO".

Line 290: Insert "of" between "because" and "low"

Line 300-302: This sentence is a fragment as is. Consider revising.

Line 338: Consider changing the word "at" to "in" or "of"

Line 345-346: This sentence is a fragment, consider revising.

Line 351: Insert "the" before "boundary layer"

Line 378: Suggest changing "to the central and southeast of Brazil" to "in central and southeast Brazil"

We agree to all these minor points and have changed the revised manuscript accordingly.

**References**

Hudman et al. (2012). Steps towards a mechanistic model of global nitric oxide emissions: implementation and space-based constraints. Atmos. Chem. Phys. 12, 7779-7795.

Liu et al. (2016). Isoprene photochemistry over the Amazon rainforest. Proc. Natl Acad. Sci. USA 113, 6125-6130.

Citation: https://doi.org/10.5194/egusphere-2025-778-RC1

**Referee #2:**

Sun et al. (2025) present a comparison of bottom-up and top-down isoprene emission estimates over tropical South America for 2019. The work is the first to use the CrIS retreivals from RAL's MLS scheme for this purpose. The model performance using updated emissions is compared to remotely sensed HCHO and in-situ isoprene measurements. The authors discuss potential drivers of biases in bottom-up emission estimates, and potential impacts on modeled ozone production sensitivities in the region.

Overall, the work is on an important topic and suitable for the audience of ACP. Additional details are needed to verify methodological soundness. Specific comments are given below.

**Major comments**

(1) More details are needed to confirm the maturity of the CrIS isoprene product used in the top-down analysis. The manuscript refers the reader to a Palmer et al (2022), and in the supplement of that paper, there is a discussion of the retrieval method; however, the uncertainties in the retrieval are not sufficiently detailed for the foundation of a top-down emissions study. For example, this manuscript states that "IMS column averages then to be lower than those derived from surface-based observations". It seems almost to be expected that IMS-based emissions would be lower than those from bottom-up emissions inventories. However, it is not clear how large these biases are. To address this concern, I suggest the authors (1) provide a quantitative assessment of measurement uncertainty (2) providing a comparison with other CrIS isoprene products, which may have more validation studies.

Please refer to our response to referee #1 about the evaluation of IMS isoprene product, the uncertainties in IMS-based emissions, as well as comparison with other CrIS isoprene products.

(2) It seems flux measurements are a more direct method of evaluating the top-down emissions estimates rather than relying on concentration measurements. Flux measurements have been reported previously for the ATTO tower (e.g., Langford et al., 2022). It may be that there were not flux measurements available in during the time period of this study- I am not sure. However, to whatever extent possible, the authors are encouraged to leverage observation-based assessments of isoprene fluxes to contextualize their work.

Thanks for the suggestion. We have now added comparison of modelled seasonal isoprene fluxes with observations. Please refer the response to referee #1. We compiled some recent flux measurements in the Amazon and compared with isoprene flux simulated with and without CrIS based isoprene emissions as shown in Table 3 above. We will include this in the revised manuscript.

(3) The GEOS-Chem model simulation requires further evaluation. Doe the model systematically misrepresent NOx concentrations and/or PBL height? If so, how sensitive are the results to any biases in these parameters? Are there other measurements (e.g., isoprene oxidation products, ozone?) that could be used to assess model performance?

Please refer to our response to referee #1 about this issue. We have evaluated modelled $NO_2$ against TROPOMI $NO_2$ and discussed impacts of model biases in NOx concentrations on isoprene lifetime and our satellite based isoprene emissions. Besides satellite retrievals of HCHO which is the major isoprene oxidation product, CO, $NO_2$ and $O_3$ can also be used to assess model performance. CO is mainly affected by wildfires and anthropogenic emissions,

similar with NO$_2$. Model biases in tropospheric O$_3$ from biases in isoprene emissions are indirect and trivial compare with other bias sources.

(4) The manuscript compares NME between TROPOMI and GEOS-Chem as a method of assessing the improvement offered by top-down emissions. Because difference in spatial patterns are also highlighted as a result, it follows that spatial correlation between GEOS-Chem and TROPOMI in the top-down and bottom-up scenarios should also be addressed within the main body of the manuscript. Table S1 in the supplement suggest the bottom-up inventory may produce a similar, and even sometimes better, correlation. How do the authors interpret this result?

Thanks for the suggestion. We have moved the Table S1 to the main body of the manuscript. As the referee pointed out: CrIS-derived emissions have unchanged correlation compared with default emissions (typo in Supplement where CrIS-inferred correlation R is supposed to be 0.78 for Jan) except for degraded correlation over May (CrIS R = 0.54; MEGAN R = 0.61) and Nov (CrIS R = 0.69; MEGAN R = 0.72). The distribution of monthly HCHO is also strongly affected by non-biogenic sources such as wildfire and anthropogenic emissions such as transportation and industrial processes. Improvement in biogenic source alone doesn't improve the overall biases significantly. As we assume zero isoprene emissions where satellite data cannot be used to derive emission rates, this potential underestimation can also increase biases.

(5) I do not find the HCHO:NO2 work to contribute meaningfully to the aims of this paper. There is a cursory acknowledgment in the problems associated with threshold-based analysis (line 340), but those caveats are not considered addressed or considered. The use of fixed numerical values (line 365) from a paper centered on China should either be removed, or rationalized more clearly.

Agreed. We have now removed this section in the revised version.

Minor and technical comments
Line 33: A lifetime of ~1 hr is given for isoprene. According to the GEOS-Chem model, what is the range in mid-day isoprene lifetime in gridcells over tropical South America?

The mid-day lifetime (<500m, ~13:00 LT) of isoprene over the tropical South America grid cells simulated with GEOS-Chem model ranges from 0.1 hr to 5 hr, with a mean lifetime of 0.5hr. Please also refer to the response to referee #1 for mid-day isoprene lifetime distributions above.

Line 97: More detail is required on the "standard" _HEMCO configuration, as the manuscript currently presumes the reader knows what is included, and "standard" _could change. For example, is CO2 inhibition or drought response included? Where it the LAI data from? Is this offline emissions using GEOS-FP meteorology?

We have added more details regarding our HEMCO configuration in the revised manuscript.
"… *The offline biogenic VOC emissions are generated with GEOS-Chem v12.3 using the Yuan processed MODIS XLAI and GEOS-FP (0.25° x 0.3125°) meteorology field. CO$_2$ inhibition is included (390 ppmv). MEGAN has a soil dependence algorithm which is disabled in GEOS-Chem. As such, our model configuration does not include any drought effects on isoprene emission. …*"

Line 139. "Although the a priori constraint on the retrieval is weak, this is also accounted for": I am not sure what "this" _refers to, what "accounted for" _means, and how that is accomplished. Please clarify.

We use a constant reference profile and our result is not influenced by the *a priori* information on the geographical distribution of likely sources and thus a weak *a priori* constraint. We still use a reference profile of 1 ppbv through all levels to account for the *a priori* information.

Line 163: For what regions was this bias-correction formula derived for? How do we know they are applicable here?

The bias correction formula was conducted over the USA, but has been evaluated against worldwide aircraft in situ HCHO concentration measurements and FTIR (Fourier-transform infrared) column observations. The corrected HCHO columns have also been evaluated for South America and that bias-corrected HCHO columns have lower biases compared with two FTIR stations in South America. We have added this information in the revised manuscript.

Line 172: Isoprene measurements from three measurement heights are used. How are these matched to GEOS-Chem levels? How is the data aggregated into the results shown in figure 3? Is there a sharp spatial gradient seen?

We have added a description of how GEOS-Chem isoprene mixing ratios are aligned with observations. The three measurement heights correspond to the first three levels from the surface. Isoprene mixing ratios from these three levels are averaged. Modelled isoprene mixing ratios are sampled at the day and time when observations are available, and then averaged over the same time period as observations. There is no sharp vertical gradient in the observations.

Line 174: Notation on the Ringsdorf citation needs to be corrected.

Corrected. "… *as described by Ringsdorf et al. (2023)* …"

Section 2.5: What regression method was used? Does it incorporate error in the x and y axis? The paragraph starting at line 194 and the first paragraph were somewhat redundant. Combining these could help with readability.

We use linear regression model (ordinary least squares, OLS) assuming the errors are normally distributed and we do not incorporate error in x or y axis. We understand there are model errors in the independent variable (isoprene emission rate, $E_{\mathrm{isop,MEGAN}}$) and in the dependent variable (isoprene column, $\Omega_{\mathrm{GC}}$) which we use to derive satellite based emission rates. We make changes in the revised manuscript to remove this redundancy.

Line 326: The third sentence of this paragraph is a fragment.

Corrected.

> "*GEOS-Chem simulated $W_{HCHO}$ depends largely on isoprene emissions over the studied region, so that any reduced biases in model HCHO can be attributed to the CrIS isoprene emission estimates.*"

**References**

Alves, E. G., Jardine, K., Tota, J., Jardine, A., Yãnez-Serrano, A. M., Karl, T., Tavares, J., Nelson, B., Gu, D., Stavrakou, T., Martin, S., Artaxo, P., Manzi, A., and Guenther, A.: Seasonality of isoprenoid emissions from a primary rainforest in central Amazonia, Atmos. Chem. Phys., 16, 3903–3925, https://doi.org/10.5194/acp-16-3903-2016, 2016.

Barkley, M. P., Palmer, P. I., Kuhn, U., Kesselmeier, J., Chance, K., Kurosu, T. P., Martin, R. V., Helmig, D., and Guenther, A.: Net ecosystem fluxes of isoprene over tropical South America inferred from Global Ozone Monitoring Experiment (GOME) observations of HCHO columns, J. Geophys. Res., 113, 2008JD009863, https://doi.org/10.1029/2008JD009863, 2008.

Bauwens, M., Stavrakou, T., Müller, J.-F., De Smedt, I., Van Roozendael, M., Van Der Werf, G. R., Wiedinmyer, C., Kaiser, J. W., Sindelarova, K., and Guenther, A.: Nine years of global hydrocarbon emissions based on source inversion of OMI formaldehyde observations, Atmos. Chem. Phys., 16, 10133–10158, https://doi.org/10.5194/acp-16-10133-2016, 2016.

Gomes Alves, E., Aquino Santana, R., Quaresma Dias-Júnior, C., Botía, S., Taylor, T., Yáñez-Serrano, A. M., Kesselmeier, J., Bourtsoukidis, E., Williams, J., Lembo Silveira De Assis, P. I., Martins, G., De Souza, R., Duvoisin Júnior, S., Guenther, A., Gu, D., Tsokankunku, A., Sörgel, M., Nelson, B., Pinto, D., Komiya, S., Martins Rosa, D., Weber, B., Barbosa, C., Robin, M., Feeley, K. J., Duque, A., Londoño Lemos, V., Contreras, M. P., Idarraga, A., López, N., Husby, C., Jestrow, B., and Cely Toro, I. M.: Intra- and interannual changes in isoprene emission from central Amazonia, Atmos. Chem. Phys., 23, 8149–8168, https://doi.org/10.5194/acp-23-8149-2023, 2023.

Hudman, R. C., Moore, N. E., Mebust, A. K., Martin, R. V., Russell, A. R., Valin, L. C., and Cohen, R. C.: Steps towards a mechanistic model of global soil nitric oxide emissions: implementation and space based-constraints, Atmos. Chem. Phys., 12, 7779–7795, https://doi.org/10.5194/acp-12-7779-2012, 2012.

Jardine, K. J., Monson, R. K., Abrell, L., Saleska, S. R., Arneth, A., Jardine, A., Ishida, F. Y., Serrano, A. M. Y., Artaxo, P., Karl, T., Fares, S., Goldstein, A., Loreto, F., and Huxman, T.: Within-plant isoprene oxidation confirmed by direct emissions of oxidation products methyl vinyl ketone and methacrolein, Global Change Biology, 18, 973–984, https://doi.org/10.1111/j.1365-2486.2011.02610.x, 2012.

Karl, T., Guenther, A., Yokelson, R. J., Greenberg, J., Potosnak, M., Blake, D. R., and Artaxo, P.: The tropical forest and fire emissions experiment: Emission, chemistry, and transport of biogenic volatile organic compounds in the lower atmosphere over Amazonia, J. Geophys. Res., 112, 2007JD008539, https://doi.org/10.1029/2007JD008539, 2007.

Kuhn, U., Andreae, M. O., Ammann, C., Araújo, A. C., Brancaleoni, E., Ciccioli, P., Dindorf, T., Frattoni, M., Gatti, L. V., Ganzeveld, L., Kruijt, B., Lelieveld, J., Lloyd, J., Meixner, F. X., Nobre, A. D., Pöschl, U., Spirig, C., Stefani, P., Thielmann, A., Valentini, R., and Kesselmeier, J.: Isoprene and monoterpene fluxes from Central Amazonian rainforest inferred from tower-based and airborne measurements, and implications on the atmospheric chemistry

and the local carbon budget, Atmos. Chem. Phys., 7, 2855–2879, https://doi.org/10.5194/acp-7-2855-2007, 2007.

Langford, B., House, E., Valach, A., Hewitt, C. N., Artaxo, P., Barkley, M. P., Brito, J., Carnell, E., Davison, B., MacKenzie, A. R., Marais, E. A., Newland, M. J., Rickard, A. R., Shaw, M. D., Yáñez-Serrano, A. M., and Nemitz, E.: Seasonality of isoprene emissions and oxidation products above the remote Amazon, Environ. Sci.: Atmos., 2, 230–240, https://doi.org/10.1039/D1EA00057H, 2022.

Palmer, P. I., Marvin, M. R., Siddans, R., Kerridge, B. J., and Moore, D. P.: Nocturnal survival of isoprene linked to formation of upper tropospheric organic aerosol, Science, 375, 562–566, https://doi.org/10.1126/science.abg4506, 2022.

Ringsdorf, A., Edtbauer, A., Vilà-Guerau De Arellano, J., Pfannerstill, E. Y., Gromov, S., Kumar, V., Pozzer, A., Wolff, S., Tsokankunku, A., Soergel, M., Sá, M. O., Araújo, A., Ditas, F., Poehlker, C., Lelieveld, J., and Williams, J.: Inferring the diurnal variability of OH radical concentrations over the Amazon from BVOC measurements, Sci Rep, 13, 14900, https://doi.org/10.1038/s41598-023-41748-4, 2023.

Sarkar, C., Guenther, A. B., Park, J.-H., Seco, R., Alves, E., Batalha, S., Santana, R., Kim, S., Smith, J., Tóta, J., and Vega, O.: PTR-TOF-MS eddy covariance measurements of isoprene and monoterpene fluxes from an eastern Amazonian rainforest, Atmos. Chem. Phys., 20, 7179–7191, https://doi.org/10.5194/acp-20-7179-2020, 2020.